# MicroRNAs from *Snellenius manilae* bracovirus regulate innate and cellular immune responses of its host *Spodoptera litura*

Cheng-Kang Tang[1], Chih-Hsuan Tsai[1], Carol-P. Wu[1], Yu-Hsien Lin [1], Sung-Chan Wei[1], Yun-Heng Lu[1], Cheng-Hsun Li[1] & Yueh-Lung Wu[1✉]

To avoid inducing immune and physiological responses in insect hosts, parasitoid wasps have developed several mechanisms to inhibit them during parasitism, including the production of venom, specialized wasp cells, and symbioses with polydnaviruses (PDVs). These mechanisms alter the host physiology to give the wasp offspring a greater chance of survival. However, the molecular mechanisms for most of these alterations remain unclear. In the present study, we applied next-generation sequencing analysis and identified several miRNAs that were encoded in the genome of *Snellenius manilae* bracovirus (SmBV), and expressed in the host larvae, *Spodoptera litura*, during parasitism. Among these miRNAs, SmBV-miR-199b-5p and SmBV-miR-2989 were found to target *domeless* and *toll-7* in the host, which are involved in the host innate immune responses. Microinjecting the inhibitors of these two miRNAs into parasitized *S. litura* larvae not only severely decreased the pupation rate of *Snellenius manilae*, but also restored the phagocytosis and encapsulation activity of the hemocytes. The results demonstrate that these two SmBV-encoded miRNAs play an important role in suppressing the immune responses of parasitized hosts. Overall, our study uncovers the functions of two SmBV-encoded miRNAs in regulating the host innate immune responses upon wasp parasitism.

[1] Department of Entomology, National Taiwan University, Taipei 106, Taiwan. ✉email: runwu@ntu.edu.tw

Parasitic wasps possess many effectors to regulate host physiological mechanisms during parasitization to ensure their offspring successfully develop in the host. These include the production of venoms, symbiosis with polydnaviruses (PDVs), and the release of teratocytes into the host body during egg hatching[1–5]. These parasitic factors help the wasp progeny to grow and to develop within the host by regulating the host immune responses, inhibiting host growth and development, affecting endocrine hormone levels, and regulating host nutrient metabolism[6–9]. PDVs are viruses belonging to the Polydnaviridae family and are divided into two genera, bracoviruses (BVs) and ichnoviruses (IVs), according to its associated host. PDV genomes are circular double-stranded DNA and are around 190–500 kb in size. They are divided into different segments and packaged into the capsid, forming viral particles harboring different gene segments[2,10–13]. The association of PDVs with wasps enables the successful growth of the wasp in its host. During wasp oviposition, PDV particles enter the host along with wasp eggs and express viral genes by adopting the host's transcription systems. These viral genes may alter the host physiology and divert energy metabolism to providing energy to the wasp larvae, subsequently inhibiting host metamorphosis[14–16], and/or the host immune systems to provide a favorable environment for the wasp progeny[6,17,18].

Several studies have demonstrated that PDVs can regulate the developmental functions of their hosts. The genome of TrIV, a PDV, contains a TrV family gene that has been demonstrated to suppress the proliferation of host cells, thus inhibiting host development[19]. In a recent study, it was found that CvBV expressed miRNAs that could inhibit ecdysone receptor (EcR) expression, resulting in delayed development of its host, Plutella xylostella[20]. PDVs have previously been shown to inhibit both the cellular and humoral immunity of the host[21]. The former includes inhibition of encapsulation and a decrease in cell adhesion that inhibits phagocytosis[22,23], both of which result in the induction of apoptosis and, in severe cases, the disruption of hemocytes. PDVs have been found to inhibit NF-κBs and to impair host innate immunity by blocking signaling pathways and inhibiting antimicrobial protein syntheses or melanization in the host[24,25].

miRNAs are small RNA molecules (~22 nucleotides) that form stem-loop structures and function as silencers, i.e., they downregulate gene expression. miRNAs may target the promoter of messenger RNAs or degrade messenger RNAs[26]. Many miRNA functions have been identified in insects and are involved in metabolism, immune responses, social behavior, and virus-host interactions[26–29]. The parasitization of Diadegma semiclausum in Plutella xylostella larvae was found to change the miRNA profile in the host[30]. Another parasitic wasp, Cotesia vestalis, and its associated bracovirus, CvBV, produces miRNAs in P. xylostella larvae to alter the host physiology and delay its growth[20]. So far, there is no evidence of PDV-produced miRNAs to regulate host immune responses.

The endoparasitic wasp Snellenius manilae (Hymenoptera: Braconidae) is highly host-specific for Noctuidae larvae[31]. S. manilae has a symbiotic relationship with a polydnavirus, S. manilae bracovirus (SmBV). In this study, we investigated whether the non-coding regions in the SmBV genome produce miRNAs along with their potential in regulating host immune functions. To this end, next-generation sequencing (NGS) was exploited to screen for miRNAs only expressed by SmBV in the host. Functional analysis of these miRNAs revealed their participation in regulating different physiological mechanisms such as growth, metabolism, and immune responses. Specifically, we found that miR-199b-5p and miR-2989 inhibited the host immune genes domeless and toll, respectively, and affected encapsulation and phagocytosis in a cellular immunity assay. Treatment using miRNA inhibitors targeting miR-199b-5p and

miR-2989 restored host immune responses and decreased the pupation rate of wasps. These results demonstrate that miRNAs produced by SmBV can inhibit the host immune responses to enable the successful growth of wasp eggs in the host.

## Results

### SmBV genome encodes abundant potential miRNAs that are associated with different physiological genes in the host.

After S. manilae had parasitized the host S. litura larva, SmBV was delivered into the host (Supplementary Fig. 1a). This resulted in the inhibition of both cellular and humoral immunity, as reflected by the decrease in phagocytosis (Supplementary Fig. 1b) and the decrease in expression of toll-7 and cecropin in the Toll pathway (Supplementary Fig. 1c). Since the expression of toll-7 was inhibited the most after 36 h of parasitism (Supplementary Fig. 1c), the small RNA expression profile in S. litura larvae was analyzed at 36 h post-parasitism using small RNA Hiseq next-generation sequencing (Supplementary Fig. 2a–c). Clean miRNA reads could be mapped to 855 known mature miRNAs in the databases, of which 422 miRNAs were only expressed after parasitism (accounting for 49.4% of the total) and 173 miRNAs were exclusively expressed in uninfected hosts (accounting for 20.2% of the total) (Fig. 1a and Supplementary Data 1). Therefore, it was hypothesized that certain miRNAs were produced when S. manilae parasitizes S. litura. The predicted targets of the miRNAs only expressed after parasitism were classified by Gene Ontology analysis. The target genes of these miRNAs are involved in diverse biological processes (Fig. 1b), located in various cellular compartments from the extracellular matrix, extracellular region part, extracellular region, to the membrane part (Fig. 1c), and have different molecular functions, including translation regulator activity, molecular transducer activity, receptor activity, and nucleic acid binding transcription factor activity (Fig. 1d and Supplementary Fig. 2d). The target genes with functions related to the host immune system were selected for analyzing the differential expression of their corresponding miRNAs. A total of 113 miRNAs were found to target immune-related genes (Supplementary Data 1) and those with a normalized $\log_2$ ratio (PSL-36/HSL-CN) higher than 10 were selected. These selected miRNAs were analyzed using miRbase to determine their known function and structure; it was found that they mostly target genes related to the innate immune pathway or cellular immunity. For instance, miR-2989 targets the upstream genes, and miR-3871-5p, miR-3552, and miR-8 affect the downstream AMP gene in the Toll pathway of innate immunities. In the JAK/STAT pathway, miR-151-5p, miR-34a-3p, miR-2808a-3p, miR-1621-5p, and miR-199b-5p target the upstream genes, miR-291a-5p simultaneously affects the upstream and downstream genes, and miR-219-3p affects the downstream AMP gene (Fig. 1e). In total, 11 miRNAs were selected for further study (Supplementary Table 1).

### SmBV-derived miR-199b-5p and miR-2989 are expressed in parasitized hosts.

To confirm the expression of the 11 miRNAs obtained through NGS in S. litura after parasitism, a stem-loop qPCR was used to compare the expression levels of the miRNAs in parasitized larvae. Most miRNA candidates presented a lower or comparable expression level to that of control let-7 miRNA; however, miR-3552 and miR-2989 targeting the drosomycin and toll genes, respectively, in the Toll pathway and miR-199b-5p targeting domeless in JAK/STAT pathway exhibited higher expression levels (Fig. 1f). To further confirm that these miRNA candidates were derived from SmBV, their expression levels in SmBV-infected S. litura cell lines (SL1A cells) were analyzed. The results showed that miR-199b-5p and miR-2989 were highly expressed at 48 h after SmBV infection, but the expression of miR-3552 was undetectable

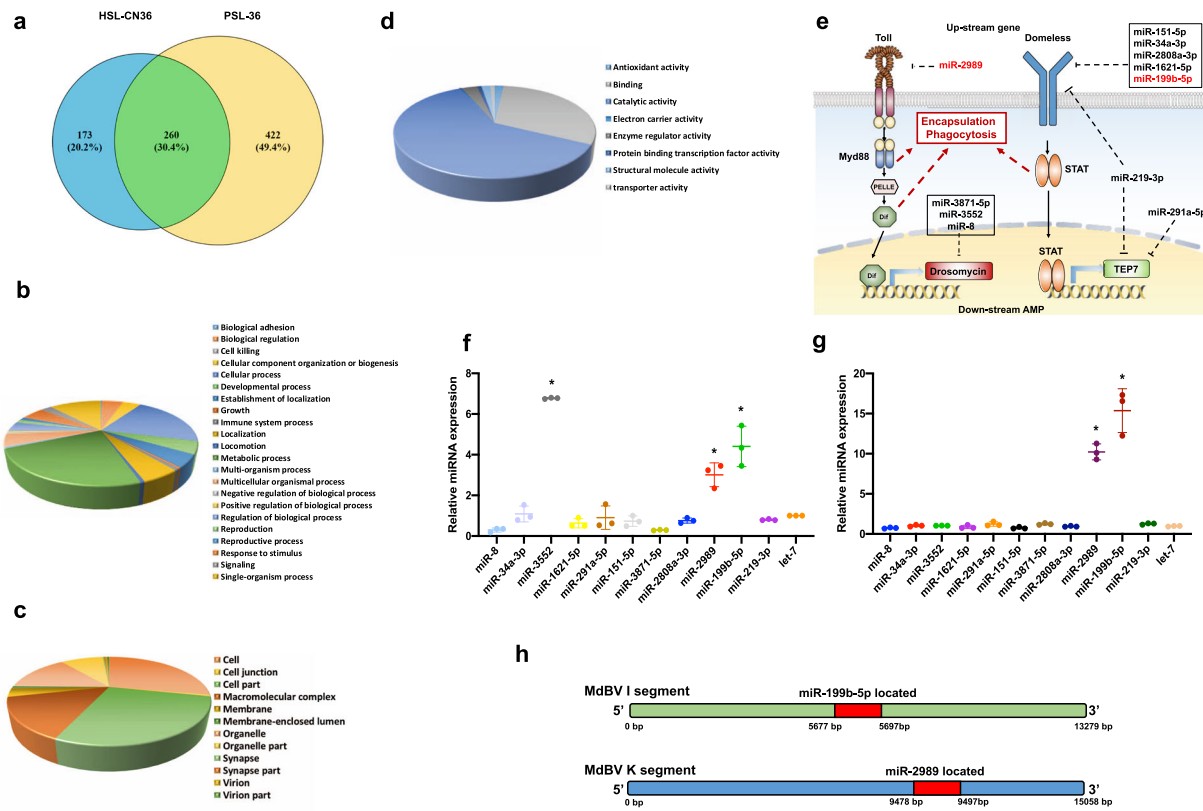

**Fig. 1 Different miRNAs expression in parasitized and unparasitized *S. litura*. a** miRNAs produced in unparasitized *S. litura* (HSL-CN) and *S. manilae*-parasitized *S. litura* (PSL-36; 36 h post-parasitism). Genes that may be affected by mature miRNAs expression in *S. manilae*-parasitized *S. litura* were identified by miRBase and classified according to their involved biological processes (**b**), cellular compartments (**c**), and molecular functions (**d**). **e** miRNAs produced in *S. litura* after *S. manilae* parasitism with gene targets predicted in the Toll and JAK/STAT pathways. **f** The expression levels of 11 predicted miRNAs were quantified by a stem-loop qPCR in *S. litura* third-instar larvae 48 h after *S. manilae* parasitism. The expression of *let-7* miRNA serves as a positive control. **g** The expression levels of 11 predicted miRNAs were quantified using stem-loop qPCR in SL1A cells 48 h after SmBV infection, with *let-7* miRNA as the positive control. A Ct value of 35 was set to be the cutoff for detection. **h** miR-199b-5p and miR-2989 are present on MdBV genome segments I and K. The red bars represent the locations of the two precursors. At least three repetitions were conducted for each group. *p*-value were calculated using Student's *t*-test (\**p* < 0.05).

(Fig. 1g). To eliminate the possibility that these miRNAs were derived from either *S. litura* or *S. manilae*, their expression levels were analyzed in the following: *S. litura* injected with purified SmBV (Supplementary Fig. 3a), female *S. manilae* alone (Supplementary Fig. 3b), and both parasitized and unparasitized *S. litura* (Supplementary Fig. 3c–d). Expression of miR-199-5p and miR-2989 was detected only in *S. litura* injected with purified SmBV or parasitized with the wasp. To further confirm the results from stem-loop PCR, a northern blot analysis was performed to detect the expression of either the mature miRNA or the precursor miRNA of miR-199b-5p and miR-2989. Our results showed that both mature and precursor miRNAs were detected in SmBV infected or parasitized *S. litura* (Supplementary Fig. 3e). However, they were not detected in non-parasitized *S. litura* (Supplementary Fig. 3e). These results indicated that the SmBV was indeed processed into miRNAs during the infection of *S. litura*. A bioinformatics analysis was also performed, and no corresponding precursor sequences were found on the genome of *S. litura*, confirming that miR-199b-5p and miR-2989 were derived from SmBV.

We mapped these two miRNAs in the genome of MdBV, a close PDV to SmBV, and found these two miRNAs were mapped to the proviral segments I and K, respectively, on the MdBV genome (Fig. 1h). These results show that miR-199b-5p and miR-2989 were derived from SmBV and were highly expressed only after SmBV entered the host.

**miR-199b-5p and miR-2989 are the main miRNAs suppressing the host immune system and target the JAK/STAT and Toll pathways, respectively.** miR-199b-5p and miR-2989 were predicted to target the genes *domeless* and *toll-7*, respectively. The expression levels of *domeless* and *toll-7* in SL1A cells infected with SmBV decreased at 48 h post infection compared to non-infected cells and cells in the initial stage of infection (Fig. 2a), suggesting that immune responses were inhibited upon SmBV infection. To confirm that miR-199b-5p and miR-2989 contributed to the decreased expression of *domeless* and *toll-7*, miR-199b-5p and miR-2989 mimics and a negative control miRNA were transfected into SL1A cells, respectively. Quantitative PCR results showed that miR-199b-5p and miR-2989 mimics inhibited the gene expression of *domeless* and *toll-7*, respectively (Fig. 2b), proving that miR-199b-5p and miR-2989 can downregulate the expression of these immune genes. To assess whether the observed regulations resulted from direct miRNA base paring, the 3′UTR of both target genes (i.e., *domeless* and *toll-7*) were fused to the 3′-end of an *egfp* gene to generate EGFP reporter plasmids pKShE-199b-5p 3′UTR and pKShE-2989 3′UTR, respectively (Fig. 2c). Co-transfection of these EGFP reporter plasmids with corresponding miRNAs decreased EGFP levels as evaluated by Western blotting analysis (Fig. 2d) and fluorescence intensity (Fig. 2e); this further confirmed that the expression of *domeless* and *toll-7* were suppressed by miR-199b-5p and miR-2989, respectively.

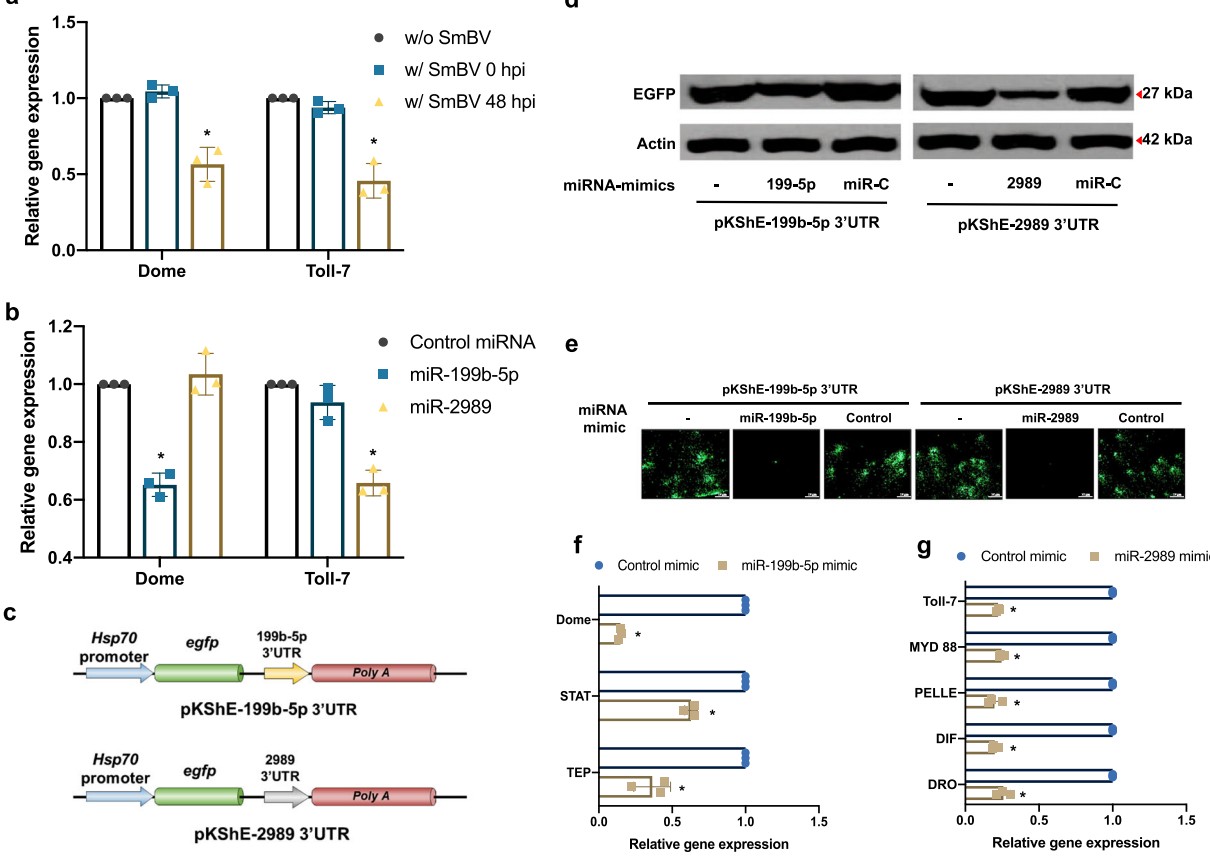

**Fig. 2 miR-199b-5p and miR-2989 inhibit the humoral immunity pathway. a** Expression levels of humoral immunity genes, *domeless* (Dome) and *toll-7* (Toll-7), were quantified by qPCR in SL1A cells 48 h after infection with SmBV. **b** Expression levels of *domeless* (Dome) and *toll-7* (Toll-7) were quantified by qPCR in SL1A cells 48 h after transfection with mimic of miR-199b-5p and miR-2989, and a control miRNA, respectively. **c** Reporter plasmids pKShE-199b-5p 3'UTR and pKShE-2989 3'UTR in which the 3'-UTR of Domeless or Toll-7 were fused to the 3'-end of an EGFP coding region, respectively. These two reporter plasmids were transfected into SL1A cells with corresponding miRNA mimic or a control miRNA (Control). EGFP expression was detected at 48 h after transfection and quantified by **d** western blotting analysis and **e** fluorescent microscopy. Genes in JAK/STAT (**f**) and Toll (**g**) pathways decreased in expression after the transfection of miR-199b-5p- or miR-2989 mimic. The expression levels of genes in the signal cascade of the JAK/STAT pathway [*domeless* (DOME), *STAT* (STAT), and *TEP* (TEP)] and the Toll pathway [*toll* (TOLL), *myd88* (MYD88), *pelle* (PELLE), *dif* (DIF), and *drosomycin* (DRO)] were determined by qPCR in SL1A cells transfected with miRNA mimics of miR-199b-5p and miR-2989, respectively. An *actin* signal was used as an internal control to normalize all readings. All experiments were performed with three biological replicates (*n* = 3). Data are expressed as the mean and standard deviation (SD). *p*-value were calculated using Student's *t*-test (\**p* < 0.05).

The effects of miR-199b-5p and miR-2989 on the expression of other important genes downstream of the JAK/STAT and Toll pathways, were also analyzed by transfecting miR-199b-5p and miR-2989 mimics into SL1A cells. Gene expression analysis result showed that the expression of three immune genes in the JAK/STAT pathway (*domeless*, *STAT*, and *TEP*) and five immune genes in the Toll pathway (*toll*, *myd88*, *pelle*, *dif*, and *drosomycin*) were decreased (Fig. 2f–g). These results demonstrate that two miRNAs derived from SmBV alter host humoral immunities by targeting upstream genes in the JAK/STAT and Toll pathways. The expression of the downstream genes was also reduced, and this may subsequently affect cellular immunities such as encapsulation and phagocytosis that are induced by the downstream gene expression.

**miR-199b-5p and miR-2989 affect phagocytosis and encapsulation in cellular immunity.** Phagocytosis and encapsulation play important roles in cellular immunity in insects. They are induced by STAT in the JAK/STAT pathway and Pelle and Dif in the Toll pathway[32]. To examine the effects of miR-199b-5p and miR-2989 on cellular immunity, miR-199b-5p inhibitor and miR-2989

inhibitor were synthesized. The specificity between miRNA mimic and its miRNA inhibitor was demonstrated by transfecting miRNA mimic and EGFP reporter plasmid into SL1A cells with or without the corresponding miRNA inhibitor. Transfection of either miR-199b-5p- or miR-2989 mimic alone decreased the EGFP level, while co-transfection of miRNA mimic and its matching inhibitor did not have an effect on the EGFP level (Supplementary Fig. 4), thus demonstrating the specificity. miR-199b-5p- or miR-2989 inhibitor were microinjected into the hemolymph of SmBV-infected third-instar *S. litura* larvae. The phagocytosis assay result showed that phagocytosis capacity was restored in insects injected with miR-199b-5p or miR-2989 inhibitor compared to insects injected with negative control inhibitor or solely with virus infection (Fig. 3a, b). Similar results were obtained using flow cytometry to measure the phagocytic hemocytes (Supplementary Fig. 5a, b). An encapsulation assay in vitro was performed to determine the binding of multiple hemocytes to foreign particles. It was found that hemocytes from insects injected with miR-199b-5p- or miR-2989 inhibitor increased the binding to Sephadex A-25 beads added into the cell culture, compared to hemocytes added with NC inhibitor or with virus infection (Fig. 3c). In vivo encapsulation assays were also

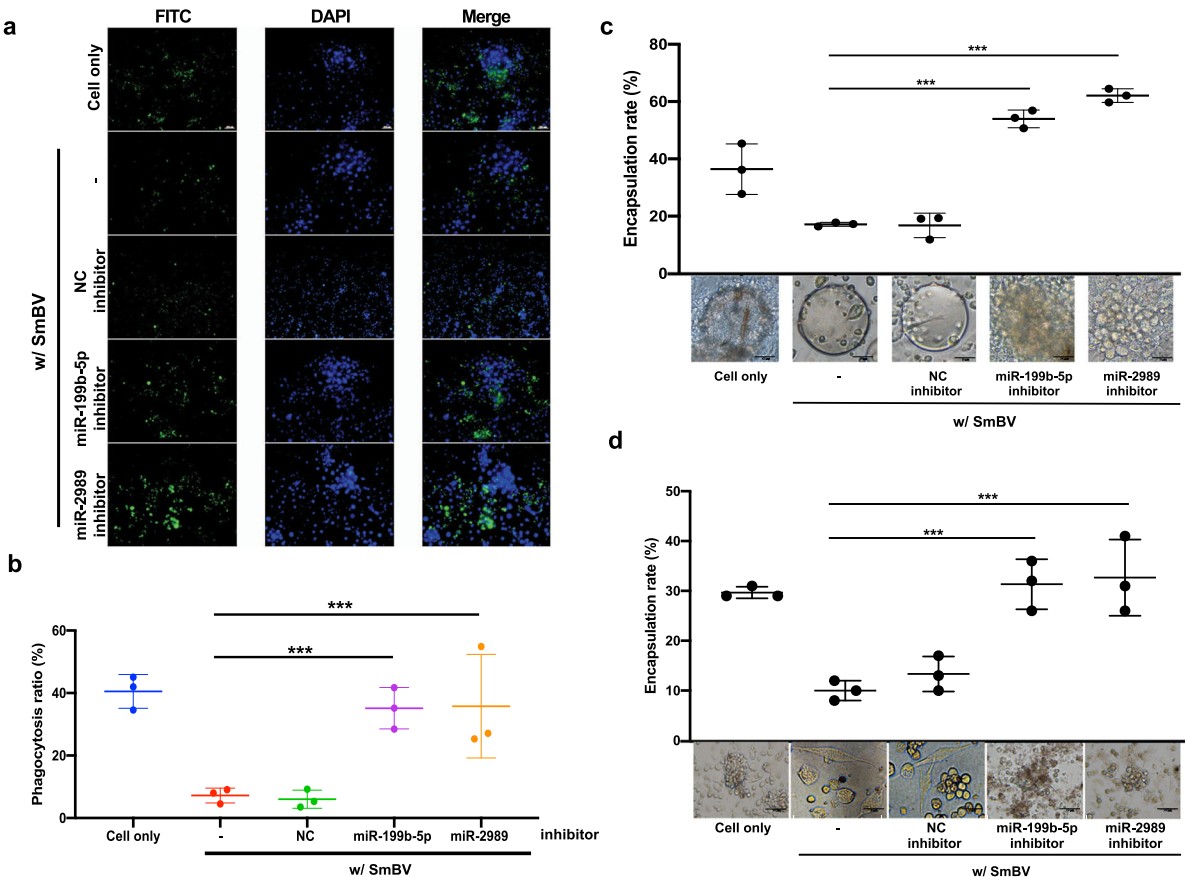

**Fig. 3 Inhibition of SmBV miRNA expression increases cellular immune responses in *S. litura*.** Third-instar *S. litura* larvae were injected with SmBV and miR-199b-5p or miR-2989 inhibitor; hemocytes were extracted 36 h after the microinjection. **a** The hemocytes were mixed with FITC-labeled *E. coli* and the phagocytosis activity of hemocytes was determined by green fluorescence emitted from the ingested *E. coli*. **b** The phagocytosis ratio (%) was derived from the proportion of FITC to DAPI; FITC can be seen as green fluorescence detected from FITC-labeled *E. coli* and DAPI can be seen as blue fluorescence detected from stained hemocytes. (The *p*-value of miR-199b-5p inhibitor: 0.00548445 and miR-2989 inhibitor: 0.04668200). **c** Encapsulation assay determining binding of multiple hemocytes to the Sephadex A-25 beads added in the cell culture. The encapsulation rate was calculated by KP assay. Bottom: representative images of one of the Sephadex A-25 beads added in each cell culture. (The *p*-value of miR-199b-5p inhibitor: 0.00078001 and miR-2989 inhibitor: 0.00019559). **d** Encapsulation of *C.elegans* in the hemocoel of *S. litura*. 24 h after injection, encapsulated nematodes were recovered from *S. litura*. (The *p*-value of miR-199b-5p inhibitor: 0.00491668 and miR-2989 inhibitor: 0.00416611) NC inhibitor: negative control inhibitor. All experiments were performed with eight biological replicates ($n = 8$). Data are expressed as the mean and standard deviation (SD). *p*-values were calculated using Student's *t*-test (*$p < 0.05$; **$p < 0.01$; ***$p < 0.0005$).

performed to determine the binding ability of hemocytes to nematodes. The nematodes were injected into *S. litura* larvae with different treatments and were flushed out from *S. litura* hemocoel 24 h after the injection. We found that nematodes from the hosts injected with miR-199b-5p- or miR-2989 inhibitor were bound by the hemocytes of the host. (Fig. 3d). These results suggest that miR-199b-5p and miR-2989 are responsible for the reduced cellular immunity in *S. litura*.

To further ensure that these two SmBV-encoded miRNAs function to suppress host innate immunity, miR-199b-5p- or miR-2989 mimic was injected into third-instar larvae of *S. litura* to assess their effects on immune gene expressions and immune responses, e.g., phagocytosis and encapsulation. Gene expression analysis result showed that the expression of immune genes in the JAK/STAT pathway and the Toll pathway were decreased after miRNA mimic injections (Supplementary Fig. 6). This, in turn, contributed to the suppressed phagocytic and encapsulation activity in insects injected with miR-199b-5p- or miR-2989 mimic (Figs. 4a, b). To further assess the effect of these two miRNA mimics on host immune responses to foreign pathogens, 24 h after the microinjection of miR-199b-5p- or miR-2989 mimic, *E.*

*coli* K-12 strain ($1 \times 10^5$ colony-forming unit (CFU)) was injected into third-instar *S. litura* larvae. The survival rate of injected insects was monitored every hour for up to 144 h (Fig. 4c). A *p*-value of less than 0.05 was considered statistically significant. Prior injection of either miR-199b-5p mimic ($p < 0.001$) or miR-2989 mimic ($p < 0.01$) significantly decreased the survival rate of *S. litura* after infection with *E. coli* K-12 as compared to *S. litura* not injected with miRNA mimic or injected with control mimic (Fig. 4c). At 144 h, the survival rate of control mimic larvae was 50%, while that of larvae injected with either miRNA mimic was less than 10%. These results demonstrate that miR-199b-5p and miR-2989 can inhibit immune responses against foreign pathogens and increase mortality from *E. coli* infection.

**Inhibitors of miR-199b-5p and miR-2989 suppress the pupation of *S. manilae*, resulting in normal growth of parasitized *S. litura* larvae.** Since the treatment of miR-199b-5p and miR-2989 inhibitors restored the cellular immunity of *S. litura*, we further examined whether the immune suppression by miRNAs influences the growth status of parasitized *S. litura* larvae and the

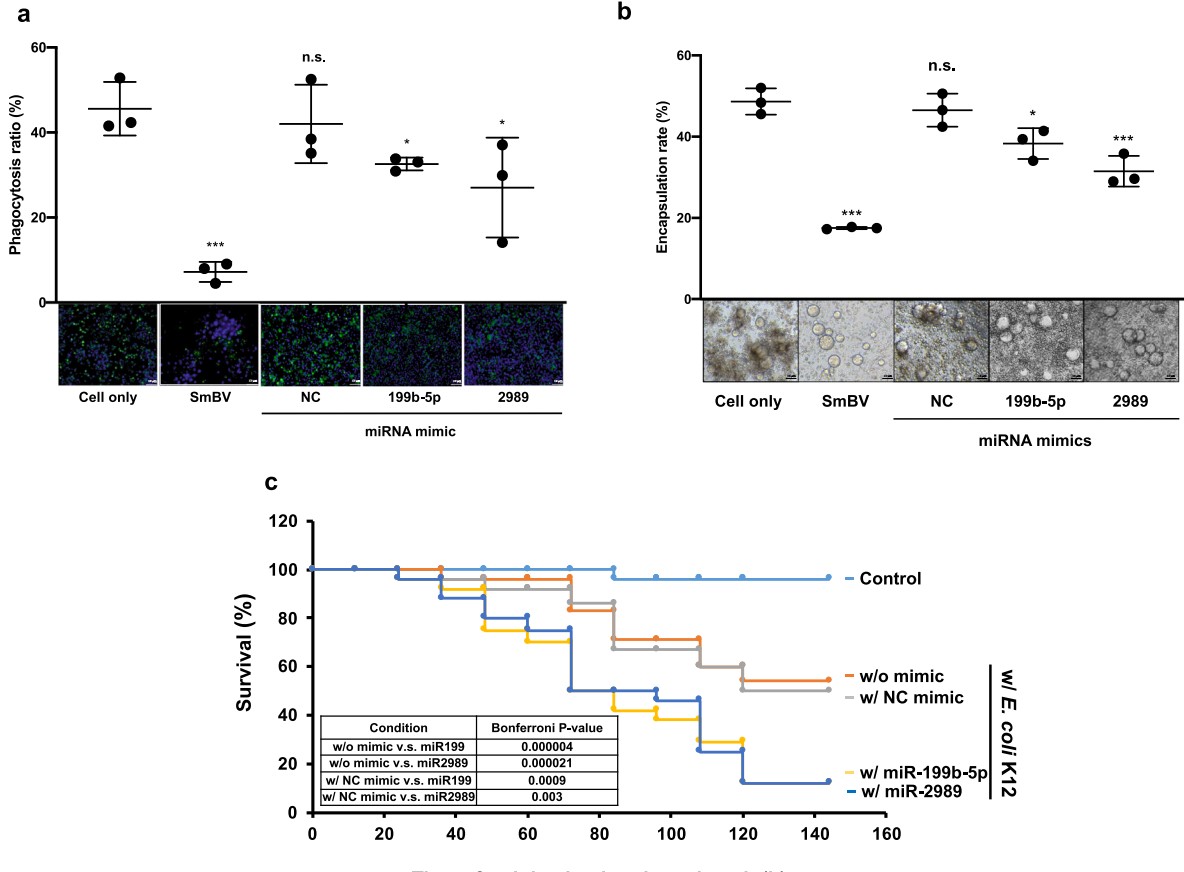

**Fig. 4 SmBV-encoded miR-199b-5p and miR-2989 suppress cellular immune responses in *S. litura*.** Third-instar *S. litura* larvae were microinjected with SmBV, miR-199b-5p or miR-2989 mimic. **a** Phagocytosis activity of injected *S. litura* larvae; green fluorescence is emitted from the ingested *E. coli* by hemocytes. The phagocytosis ratio (%) was derived from the ratio of FITC to DAPI. (The *p*-value of SmBV: 0.00210683, NC mimic: 0.30700539, 199b-5p mimic: 0.03159918, 2989 mimic: 0.04655791). **b** Encapsulation assay showing binding of multiple hemocytes to Sephadex A-25 beads added to the cell culture. The encapsulation rate was calculated by KP assay. Bottom: representative images of the Sephadex A-25 beads added to each cell culture. (The *p*-value of SmBV: 0.00168336, NC mimic: 0.25959650, 199b-5p mimic: 0.01188009, 2989 mimic: 0.00211812). **c** Survival rate of larvae in response to infection by *E. coli* K12. Kaplan–Meier survival curve with log-rank test (Wilcoxon–Breslow–Gehan Test) comparing survival of *S. litura* larva infected with *E. coli* K12. A *p*-value of less than 0.05 was considered statistically significant. Pairwise comparison: with NC mimic vs. with miR-199b-5p, *p* < 0.0001; with NC mimic vs. with miR-2989, *p* < 0.01. NC mimic: negative control mimic. All experiments were performed with five biological replicates (*n* = 5). Data are expressed as the mean and standard deviation (SD). *p*-value were calculated using Student's t-test (**p* < 0.05; ***p* < 0.01; ****p* < 0.0005).

successful pupation of *S. manilae* in *S. litura*. The determination of body length and head capsule width of *S. litura* showed that control larvae without any treatment presented normal development, whereas parasitized larvae without treating with miRNA inhibitor exhibited arrested development (Fig. 5a–c). Notably, the parasitized larvae injected with each miRNA inhibitor presented normal development, in contrast to the parasitized larvae injected with inhibitor control (Fig. 5a–c). In addition, treatments of miRNA inhibitor in parasitized *S. litura* larvae suppressed the development of *S. manilae* larvae. The pupation rates of *S. manilae* larvae in miRNA inhibitor treatment groups were lower than 20% in comparison to the nearly 95% pupation rate of control groups (Fig. 5d). The encapsulation activity of *S. manilae* egg and larvae was higher in both miRNA inhibitor treatment groups than in the control groups (SmBV infected or SmBV infected with inhibitor control) (Fig. 5e). These results confirmed that miRNA inhibitor treatments recovered the host cellular immune responses, which obstruct the development of *S. manilae* larvae. To further ensure that both SmBV-encoded miRNAs suppress host innate immunity rather than larval developmental pathways, we injected each miRNA mimic into second-instar larvae of *S. litura* and assessed their development. The results

showed that neither miRNAs could suppress development (Fig. 5f–h). In conclusion, our results demonstrated that inhibiting the actions of both SmBV-encoded miRNAs by miRNA inhibitors restored the host immune responses, thus suppressing *S. manilae* development in the host.

## Discussion

PDVs have been shown to assist parasitoid wasps in successfully parasitizing their hosts. PDVs produce protein products that can directly block or compete for receptors in the physiological and metabolic pathways of the host, causing delayed growth and arrested development[18,33]. PDVs also express viral proteins to disrupt the adhesion and dissemination of immune cells, which disable phagocytosis and encapsulation in hemolymph[18,22,23,34–36]. Furthermore, PDVs inhibit melanization and humoral immunity pathways, resulting in the inhibition of AMP synthesis by the host[3,21,25,37]. In addition to viral genes and proteins affecting host functions, non-coding genes of viruses have also been shown to play important regulatory roles. Recently, miRNAs produced by PDVs were shown to delay the growth of the host[20]. In this study, we demonstrated that miRNAs encoded by the non-coding regions

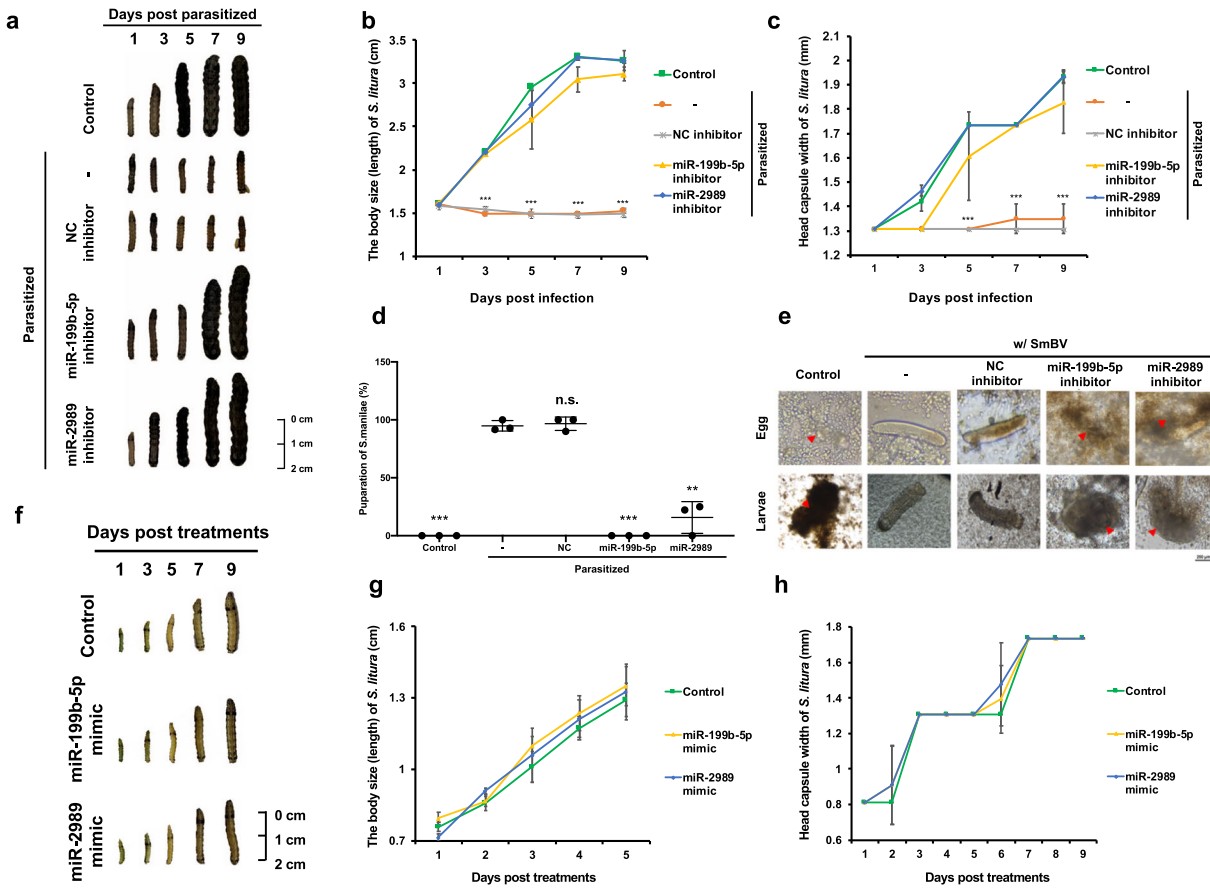

**Fig. 5 Inhibition of SmBV miRNA expression decreases *S. manilae* parasitism rate. a** Third-instar *S. litura* larvae were microinjected with miRNA inhibitors before parasitism by *S. manilae* wasps for 48 h. Arrow: *S. manilae*. The developments of *S. litura* larvae were measured by their **b** body sizes (lengths) and **c** head capsule width after parasitism (Control: unparasitized *S. litura*) (Green square: Control, orange circle: Parasitized, gray star: Parasitized + NC inhibitor, yellow triangle: Parasitized + miR-199b-5p inhibitor, blue diamond: Parasitized + miR-2989 inhibitor). (The *p*-value in **b** on 3–9 days post infection are 0.00000635, 0.00000006, 0.00001210, and 0.00000030) (The *p*-value in **c** on 5–9 days post infection are 4.79293E−28, 0.00020260, and 0.00038115). **d** The pupation rate of *S. manilae* wasps in *S. litura* (*n* = 3) over 10 continuous days. The number of wasps that successfully pupated in the group without miRNA inhibitor injection was set as 100%. (The *p*-value of Control: 0.00075194, miR-199b-5p inhibitor: 0.00075194, and miR-2989 inhibitor: 0.00561026). **e** Encapsulation assay determining binding of multiple hemocytes to *S. manilae* eggs (upper panel) and 4-day-old *S. manilae* larvae (lower panel) added in the cell culture. Arrow: encapsulated and melanized *S. manilae* egg or larvae. **f** Second-instar *S. litura* larvae were microinjected with miRNA mimics and monitored over a 9 days period post-injection. (Control: *S. litura* without miRNA mimic injection.) The developments of *S. litura* larvae were measured by their **g** body sizes (lengths) and **h** head capsule width after miRNA mimic injection (Green square: Control, yellow triangle: miR-199b-5p mimic, blue diamond: miR-2989 mimic). All experiments were performed with three biological replicates. Data are expressed as the mean and standard deviation (SD). *p*-value were calculated using Student's t-test (\*\*\**p* < 0.005).

of SmBV play key roles in the suppression of the host's immune systems, which eventually benefits the pupation of *S. manilae*.

At present, approximately 300 types of viruses are known to produce miRNAs, including herpes simplex virus, baculovirus, adenovirus, and nudivirus[38,39]. These viral miRNAs play key roles in virus-host interactions[40,41] as they can directly change host physiology and interfere with host defense mechanisms[26,29,42]. For instance, viral miRNAs KUN-miR-1 and OvHV-2-miR-5 can affect host gene expression and up-regulate virus infectivity and replication in the host[38]. Previous studies have noted the changes in the expression levels of some miRNAs before and after the parasitism of Lepidoptera larvae by parasitic wasps; however, the functions of these miRNAs remained unclear[30,43]. In this study, miRNAs that are specifically expressed in *S. litura* after parasitism by *S. manilae* and SmBV were identified by small RNA Hiseq NGS. Those that were involved in immune processes were further analyzed. It was found that the expression levels of miRNAs inhibiting immune responses were higher than those involved in other processes, suggesting that

viruses may suppress host immunity and enable wasps to evade the host immune responses. In SL1A cells parasitized by wasps or directly infected by SmBV, the miRNAs miR-199b-5p and miR-2989 were both found to downregulate immune response pathways and subsequently suppress the immune systems of *S. litura*. These two miRNAs targeted the Domeless in the JAK/STAT pathway and Toll receptor in the Toll pathway, indicating that PDVs can transcribe miRNAs to regulate the different routes of host immune responses. It has previously been reported that IEP1 and IEP2 produced by CkBV and CpBV-lectin molecule produced by CpBV interfered with the recognition and adhesion of wasp eggs by immune cells (granulocytes, plasmatocytes, and oenocytoid cells) in the host hemolymph[44,45]. Additionally, MdBV has been found to produce GLC, PTP, and ANK proteins to decrease immune cell dissemination and adhesion[13,23,24,46–49]. In the present study, impaired phagocytosis and encapsulation from hosts infected with SmBV was observed (Fig. 3). Injection of miR-199b-5p and miR-2989 inhibitors into host hemolymph restored phagocytosis and encapsulation (Fig. 3b, c), thus proving

that miR-199b-5p and miR-2989 are indeed the suppressors of these host immune responses.

During induction of encapsulation, Dif in the Toll pathway induces NF-κB to activate encapsulation, whereas STAT in the JAK/STAT pathway increases the encapsulation signals. Thoet-kiattikul et al. reported that a PDV expressed two inhibitor kappa B (IκB)-like proteins, H4 and H5, from the *ANK* gene family to inhibit the expression of Dif in the Toll pathway and Relish in the Imd pathway[25], eventually suppressing the encapsulation activity and humoral immunities related to NF-κB[17,50]. This is similar to the findings presented in the present study, namely that the products expressed by PDVs effectively suppress the insect immune system and suggests that PDVs possess mechanisms to alter the host immune system to provide a survival advantage. The present study identified miRNAs (miR-199b-5p and miR-2989) transcribed by an SmBV which can directly alter humoral immunity by decreasing the expression of immune genes. miR-199b-5p and miR-2989 produced by SmBV not only suppressed humoral immunity but also indirectly blocked STAT and Dif signal transduction, decreasing phagocytosis and encapsulation (Fig. 6). To our knowledge, this is the first report showing that miRNAs derived from a PDV can target the host immune system, and that this immune suppression benefitted the parasitism of *S. manilae* larvae within the *S. litura* host.

## Methods

**Insects.** *Spodoptera litura* were reared in cages and kept in growth chambers at 29 ± 1 °C with a photoperiod of 12:12 h (light: dark). The main components of the artificial diet consisted of 90 g kidney bean powder, 36 g yeast extract, and 33 g wheat germ blended first with 480 ml of distilled water, and then with an additional 60 ml of nutrition mix (0.36g L-cysteine, 3.6 g L-ascorbic acid, and 0.225 g strep-tomycin sulfate). We then mixed 9.9 g agar powder and 300 ml distilled water in a 1000 ml beaker to form an agar mixture that was dissolved and boiled in a microwave. The agar mixture was allowed to cool down to 70 °C before it was added to the artificial diet mixture. Finally, 3 ml propionic acid was added to the artificial diet mixture to prevent mold growth. The mixture was then poured into a plastic tank to cool, and thereafter stored in a 4 °C refrigerator. *S. manilae* were reared in growth chambers at 29 ± 1 °C with a photoperiod of 12:12 h (light: dark). Glucose solution (15%) was used for feeding[51].

**Cells, SmBV particles extraction, and viral copy number determination.** *Spodoptera litura* SL1A cells were cultured in TC-100 medium (USBio) containing 10% FBS (Gibco BRL) and cultured in a 26 °C incubator. SmBV virions were collected from the ovaries of female *S. manilae* wasps, as previously described[3]. The ovaries were dissected in pre-chilled phosphate-buffered saline (PBS), and the calyx was punctured to release the content into the PBS. The solution was filtered through a 0.45-μm filter, and the filtrate was centrifuged at 20,000 × *g* for 1 h[52]. After centrifugation, the resulting pellet containing the virus was re-suspended in PBS. A plasmid containing the C fragment of SmBV was constructed to serve as the standard in qPCR to quantify the genome copy number of SmBV. The

concentration of stock plasmid DNA was measured using a NanoDrop 2000 spectrophotometer (Thermo Fisher Scientific) and the amount of DNA sample was determined to be equivalent to $10^{10}$ plasmid copy number using the DNA Copy Number and Dilution Calculator (Thermo Fisher Scientific). Ten-fold serial dilution of the DNA sample ranging from $10^{10}$ to $10^{1}$ plasmid copy number was performed, and the serially diluted samples were used as templates in qPCR. A standard curve was obtained by plotting diluted template DNA to the corresponding *Ct* value[53]. To infect *S. litura* larvae, SmBV solution containing $10^{6}$ virus copy number was injected into second-instar larvae. The expression level of SmBV-encoded miRNAs was analyzed 36 h post-injection by stem-loop qPCR.

**Small RNA Hiseq next-generation sequencing.** We collected a total of 20 second-instar *S. litura*: 10 were parasitized by *S. manilae* and 10 were unparasitized. Total RNA was extracted using Trizol reagent (Geneaid), and small RNAs (~18–30 nt) were isolated by a denaturing polyacrylamide gel electrophoresis (PAGE) as per the method of Lagos-Quintana et al.[20,54,55]. Small RNA sequencing libraries were constructed using a TruSeq Small RNA Library Preparation Kit (Illumina) and sequenced using an Illumina HiSeq 2000/2500. The obtained sequences were aligned with various known bioinformatics databases to classify and annotate the small RNAs. Non-annotated small RNAs were further analyzed by MIREAP and miRDeep to predict potential new miRNAs. Target genes were predicted for miRNAs using RNAhybrid, miRanda, TargetScan, and PITA. We analyzed their differential expression and predicted the loci of these target genes. Additionally, we analyzed the gene functions and pathways involved using Gene Ontology functional annotation and KEGG pathway annotation.

**Stem-loop qPCR.** miRNA cDNAs were synthesized from total RNA using miRNA-specific stem-loop primers (Supplementary Table 2) according to previously described criteria[56]. Expression of the mature miRNA was analyzed by qPCR as described previously[57]. Briefly, cDNA was diluted 10 times and added into a qPCR reaction mixture with a final volume of 20 μL, containing 2× SYBR Green qPCR Master Mix (Fermentas), 0.5 μM each of forward and reverse primers, 1 μL of cDNA, and 7 μL of ddH$_2$O. The qPCR was performed with the following program: 94 °C for 15 s, 60 °C for 30 s, and 72 °C for 30 s. Expression of let-7a, a miRNA normally expressed in most insect cells, was used as a positive control.

**Total RNA extraction, cDNA synthesis, and qPCR.** The total RNA of miRNA mimic-injected *S. litura* or miRNA mimic transfected SL1A cells were extracted using Trizol reagent (Geneaid)[58]. Samples were homogenized with 1000 μL of Trizol reagent, then kept at room temperature for 5 min. 200 μL of chloroform was added into homogenized samples and vortex for 15 s. After incubated for 3 min at room temperature, the samples were centrifuged at 12,000 × *g* for 15 min at 4 °C. The aqueous phase was transferred into a new eppendorf, then added 500 μL of 100% isopropanol and incubated for 10 min at room temperature. The RNA pellet was spin down at 12,000 × *g* for 15 min at 4 °C and washed by 1 mL of 75% ethanol two times at 7500 × *g* for 5 min at 4 °C. Finally, RNA pellet was dried at SpinVac at 45 °C for 3 min and dissolved in 50 μL of RNAse-free water. The concentration of the extracted RNA was determined using the Nanodrop 2000. cDNAs were synthesized using a PrimeScript™ RT Reagent Kit (Takara) with the following reaction components: 500 ng of RNA, 2 μL of 5× PrimerScript™ buffer, 0.5 μL of Primer-Script RT Enzyme Mix I, 0.5 μL of Oligo dT primer, and 0.5 μL of random hex-amers. Quantification of the expression of miRNA target genes was performed using SYBR green (Bioline)-based qPCR on an ABI Plus One real-time system (StepOnePlus, Applied Biosystems). Each qPCR reaction contained 10 μL of SYBR green reagent, 0.4 μM each of forward and reverse primers, 2 μL of cDNA, and 6.4 μL of ddH$_2$O. The qPCR cycling conditions were as follows: stage 1, 95 °C for

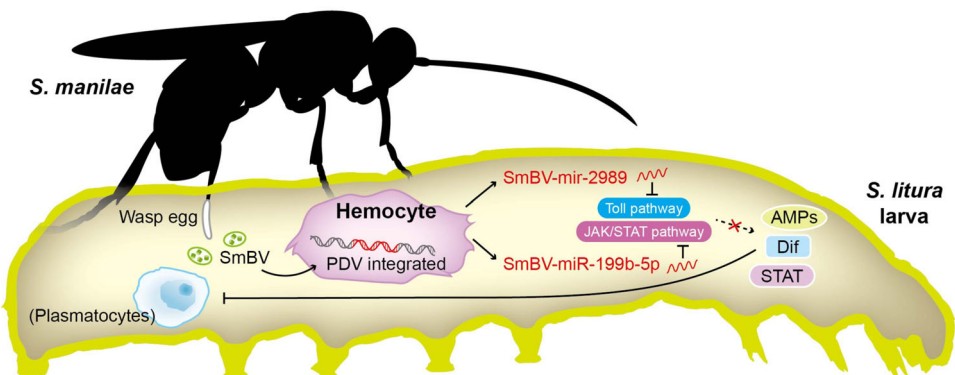

**Fig. 6 SmBV-encoded microRNAs as regulators in host immune responses.** Upon the parasitism of wasp *S. manilae* to its host *S. litura*, the symbiotic SmBV integrates viral DNA into the host genome in hemocytes, and subsequently expresses miR-199b-5p and miR-2989 that inhibit signal transductions in JAK/STAT and Toll pathways, respectively. These suppressions indirectly block the function of STAT and Dif, alongside AMP synthesis inhibition, resulting in decreased immunity in plasmatocytes and increased pupation rate of the wasp larvae in the host.

10 min; stage 2, 40 cycles of 95 °C for 15 s and 60 °C for 1 min; stage 3, 95 °C for 15 s. A list of primer sequences used in this study is given in Supplementary Table 3.

**Plasmid construction.** EGFP reporter plasmids pKShE-199b-5p 3′UTR and pKShE-2989 3′UTR were constructed using the plasmid pKShE in which an EGFP gene is driven by the *hsp 70* promoter in the backbone of pBluescript II KS(−) (Stratagene)[59,60]. The 3′UTRs of *domeless* and *toll-7* were amplified by PCR from the cDNA of *S. litura* and inserted downstream to *egfp* in the plasmid pKShE to generate the plasmids pKShE-199b-5p 3′UTR and pKShE-2989 3′UTR, respectively. The primer sets and 5′-GGTTAGTACCTATATGTGAATA-3′ and 5′-AGC ACGTCTATTGAATGTTGCTGTC-3′ were used for the 3′UTR of *domeless* and 5′-AGAGGCGTGGGGCCAAACCGAC-3′ and 5′-CACAAATTCCCCCCTCGGC AA-3′ were used to amplify the 3′ UTR of *toll-7*.

**DNA transfection and fluorescence quantification.** SL1A cells ($2 \times 10^5$ cells per well) were seeded per well in 24-well plates (Corning). 0.25 µg of reporter plasmids pKShE-199b-5p 3′UTR or pKShE-2989 3′UTR were transfected alone or with 10 pmol of the corresponding miRNA mimic or control miRNA (Control) using Cellfectin (Invitrogen) according to the manufacturer's protocol (Gibco BRL). EGFP expression was detected at 48 h after transfection by a fluorometer, and the images of cells were captured by a fluorescent microscope.

**Western blotting analysis.** After reporter plasmid and miRNA mimic (10 pmol) transfection, SL1A cells ($2 \times 10^5$) were placed in the wells of a 24-well plate, washed twice with PBS buffer, and lyzed in RIPA lysis buffer (100 µL). Sodium dodecyl sulfate (SDS) sample buffer was mixed with the cell samples in a 1:4 ratio. The mixture was then heated to 100 °C for 10 min and then separated on a 10% SDS-polyacrylamide gel. Proteins on the gel were transferred onto a PVDF membrane (Millipore) by electroblotting for 1 h in transfer buffer. The PVDF membrane was blocked with PBS containing 5% skim milk, 0.05% Tween 20 for 1 h and incubated with primary mouse anti-GFP (Millipore) or anti-Actin (Millipore) antibodies for another 1 h. The membrane was washed three times with PBS containing 0.05% Tween 20 (PBST) and then incubated with secondary antibody conjugated with horseradish peroxidase (HRP) for 1 h. After three washes with PBST, the membrane was rinsed with enhanced chemiluminescence (ECL) substrate and the luminescence signal was detected using X-ray film exposure.

**Injection of miRNA inhibitor or miRNA mimic in *S. litura* larvae.** Two microliters of miRNA inhibitor (10 pmol) was administered to third-instar *S. litura* larvae by microinjection (80 larvae per treatment). Then, each larva was infected with 10 *S. manilae* for 48 h to establish parasitism, after which time the *S. manilae* were removed. The number of *S. manilae* that successfully pupated was calculated; this was considered to be the viability of *S. manilae* under phagocytosis and encapsulation. The body length and head capsule width of hatched larva were also measured. miRNA mimic (10 pmol) was microinjected into second-instar *S. litura* larvae (20 larvae per treatment) for assay development, phagocytosis and encapsulation. Body lengths and head capsule widths of hatched larva were measured over a 9 days period post-injection. All miRNAs mimic or miRNA inhibitor were synthesized by MDbio Inc. The miRNAs or mutant miRNAs used in this study are as follows:

SmBV-miR-199b-5p (5′-CCCAGUGUUUAGACUAUCUGUUCtt-3′); SmBV-miR-2989 (5′-GCACGUGAUGAGAACUCUGUtt-3′); miR-Control (5′-UUCUCC GAACGUGUCACGU-3′); SmBV-miR-199b-5p inhibitor (5′-GAACAGAUAGU CUAAACACUGGG-3′); SmBV-miR-2989 inhibitor (5′-ACAGAGUUCUCAUC ACGUGC-3′); miR-Control inhibitor (5′-ACGUGACACGUUCGGAGAA-3′).

**Phagocytosis assay.** Third-instar larvae were injected with SmBV ($1 \times 10^6$ virus copy number). Hemocytes were extracted 36 h post-injection and seeded onto a 96-well plate ($4 \times 10^4$ cells/well). FITC-labeled *E. coli* ($4 \times 10^4$ cells/well) (BioParticles® Fluorescent Particles and Opsonizing Reagents, Molecular Probe) then added. TC-100 culture medium was added to a total volume of 100 µL. The cells were cultured at 25 °C for 60 min, followed by 2–3 washes using 10% PBS. Fifty microliters of 0.4% trypan blue was added, and cells were stained for 10 min. After 2–3 washes with 10% PBS, the cells were fixed with 4% formaldehyde and incubated for 30 s. Fifty microliters of DAPI (300 µM; Thermo Fisher Scientific) was added to stain for 30 min, followed by washing with 10% PBS 2–3 times. The percentage of FITC-labeled *E. coli* and hemocytes were observed using fluorescence microscopy (Nikon Inverted Microscope-Eclipse TS100)[35].

**Encapsulation assay.** In order to observe A-25 bead encapsulation, hemolymph was collected from third-instar larvae injected with SmBV ($1 \times 10^6$ virus copy number). Hemocytes were extracted 48 h post-injection and mixed with 1 mL of Pringle's Saline (1.541 M NaCl, 0.027 M KCl, 0.14 M CaCl₂, 0.222 M glucose) before centrifugation at 4 °C and $500 \times g$ for 5 min. The supernatant was discarded, and hemocytes were suspended in 1 mL of Pringle's Saline. After re-centrifugation, 100 µL of Pringle's Saline was used to resuspend the pellet, and the hemocyte density was calculated by a hemocytometer. Fifty microliters of hemocytes were mixed with 50 Sephadex A-25 beads into individual wells of a 96-well plate. To

observe *S. manilae* larvae encapsulation, we allowed *S. manilae* to parasitize second-instar *S. litura* and dissected them after 4 days to retrieve the *S. manilae* larvae. Following two to three washes in PBS, 50 µL of hemocytes was mixed with *S. manilae* larvae and added to individual wells of a 96-well plate. The 96-well plate was then sealed with Parafilm (M® All-Purpose Laboratory Film Bemis) and cultured at 26 °C for 30 min. Encapsulation was observed by a microscope (Nikon Inverted Microscope-Eclipse TS100) after incubation. Using Nikon NIS auto-measurement software, encapsulated objects were counted at 400x magnification. Each experimental sample was counted in three different fields under the microscope, and the data was presented as the average number of encapsulated objects from these 3 fields.

**Encapsulation of nematodes in vivo.** An aliquot of nematode (strain: N2 Bristol) suspension was autoclaved (121 °C, 1.2 atm, 20 min) to eliminate possible interference from symbiosed *E. coli*. Approximately 500 dead *C. elegans* L1 larvae in sterile M9 buffer were injected into a third-instar *S. litura* larva using a 1 mL tuberculin syringe with a 26G × ½″ needle[61]. *S. litura* larva were dissected along the centerline 24 h after injection and nematodes in the hemocoel were flushed out using phosphate-buffered saline. The obtained nematodes were observed under a microscope and were counted as either unencapsulated or encapsulated.

**Bacterial injection.** Third-instar larvae were cold-anesthetized prior to injection. A fine glass needle was inserted into the dorsal vessel and 1 µL (containing $1 \times 10^5$ colony-forming unit (CFU)) of cultured bacteria was injected into the hemocoel[62]. For survival curve, 150 larvae were divided into five groups: NC mimic-injected with *E. coli* K12, miR-199b-5p mimic-injected with *E. coli* K12, miR-2989 mimic-injected with *E. coli* K12, *E. coli* K12-injected only group, and an PBS medium-injected group (Control). The survival rate of experimental larvae in each group was recorded every 12 h until 144 h post infection. Statistical tests were performed using the online tool OASIS 2, and weighted log-rank test (Wilcoxon–Breslow–Gehan test) for determining significance[63].

**Statistics and reproducibility.** The relative expression levels of RNAs or miRNAs were calculated using the $2^{-\Delta\Delta Ct}$ formula in which *Ct* is the cycle threshold value of qPCR, and the expression of *18S* was used as a reference gene. Student's t-test was used to compare each treatment group with the control group. Groups with significant differences *versus* the control group are marked with an asterisk (*) (*$p < 0.05$; **$p < 0.01$; ***$p < 0.005$) in Figs. 1, 5, including gene expression, phagocytosis assay, encapsulation assay, and pupation test.

**Reporting summary.** Further information on research design is available in the Nature Research Reporting Summary linked to this article.

## Data availability
Source data for the NGS and those underlying plots shown in figures are provided in Supplementary Data 1 and 2. Full blots are shown in Supplementary Information. All other data generated during and/or analyzed during the current study are available from the corresponding author on reasonable request.

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

## Acknowledgements

We thank Mr. Alexander Barton for kindly revising the manuscript. This research was funded by the grant MOST107-2311-B-002-024-MY3 to Y.L.W. from the Ministry of Science and Technology, Taiwan.

## Author contributions

Guarantors of the integrity of the entire study, study concepts, and manuscript preparation: C.K.T., C.H.T., Y.H.L., and Y.L.W. Study design, data acquisition/analysis, literature research, and manuscript preparation: C.K.T., C.H.T., C.P.W., S.C.W., C.H.Lu, and Y.L.W. Data acquisition/analysis, manuscript editing, and revision: C.K.T., C.H.T., C.P.W., Y.H.Lin, and Y.L.W. All authors reviewed the manuscript.

## Competing interests

The authors declare no competing interests.
