## [Peer Review File · Communications Biology]

Reviewers' comments:

Reviewer #1 (Remarks to the Author):

Comments for COMMSBIO-19-1955-T

In the manuscript, "Snellenius manilae bracovirus-encoded microRNAs as regulators in host immune responses", Cheng-Kang Tang and co-workers proposed that Snellenius manila bracovirus-edcoded microRNAs (miRNAs), miR-2989 and miR-199b-5p, inhibited host immune responses via regulating host Toll and JAK/STAT pathways. The authors used Small RNA Hiseq Next Generation Sequencing combined with physiological assays to detect the functions of the two miRNAs. In general, the paper is interesting but the writing of this manuscript is not clear. Especially in the abstract, it didn't even contain the important results of this paper. Thus, I believe this manuscript is not ready for publish. Some suggestions are listed below for the authors' consideration.

1. The authors should combine Fig. 1 and 2 to make the paper more logical such that it shows the importance of the two miRNAs and the way to discover them.
2. It would be better to use Head Capsule Width to describe the inhibition of miRNA in Fig.5A and please also check whether the legend, "CN: S. manilae parasitized", is correct.
3. It would be helpful to add the methods used in this research to allow reviewers to understand the assays in this manuscript. For example, in Fig.3, how did the authors generate pKShE plasmid and perform western blot.
4. The authors should read this manuscript carefully to correct some wrong descriptions.

Reviewer #2 (Remarks to the Author):

The manuscript by Tang et al. describes microRNAs from Snellenius manilae bracovirus (SmBV) act as regulators in host immune responses. By comparing the miRNAs from Spodoptera litura larvae parasitized by S. manila with those unparasitized, the authors found that the SmBV encoded miRNAs into the host larvae during parasitism. Two of these miRNAs regulated the host's immune responses by blocking the expression of several key genes in the signaling pathways. Further, the authors microinjected the inhibitors of these two miRNAs into the parasitized host larvae, which rescued their immune responses and significantly decreased the S. manila pupation rate. The authors hope to show SmBV-derived miRNAs play an important role in regulating host immune responses in this study. However, a number of methodological issues existed in this paper, and consequently the work is not convincing. In places, the results are also overinterpreted, for example, the conclusion "Overall, we demonstrate a cross-species regulation by miRNAs in animal parasitism" in the abstract is misleading because cross-species regulation via miRNAs in animal parasitism was already discovered two years ago. I thus did not recommend acceptance of the manuscript at the present form.

Major concerns:

>>Line 91. SmBV genome encodes abundant 91 potential miRNAs that are associated with different physiological genes in the host..."

The authors misinterpreted "the miRNAs only expressed after parasitism" the same as "SmBV-derived miRNAs". From the experiments and the method descriptions, it is very likely that the small RNA library of the parasitized S. litura contains miRNAs from the wasps, not only from the SmBV. Furthermore, some host miRNAs were possibly up-regulated due to SmBV viral genes or other parasitism-associated factors. Therefore it is extremely important for the authors to do additional analyses or experiments to distinguish the SmBV-derived miRNAs, wasp-derived miRNAs, and host-derived miRNAs.

>>Line 102. A total of 855 known miRNAs...

The total numbers of miRNAs provided in the text are unrealistically high. No information is provided about their count numbers and pre-miRNA structures.

>>Line 142-143. These results show that miR-2989 and miR-199b-5p are present in SmBV and are highly expressed after SmBV enters the host..."

Most miRNAs are very conserved across different species. Have the authors identified the existence of these two miRNAs (miR-2989 and miR-199b-5p) in the genome of the host *S. litura*? If the host genome also expresses these two miRNAs or their homologous, the conclusion will be completely changed. Again, in the method section, the experiment to extract SmBV is too simple to exclude the wasp ovary contents, which may lead to highly expression of these two miRNAs. Thus, the fact that the two miRNAs were highly expressed here is doubtful.

>>Line 176. These results indicate that miRNAs produced by SmBV affect the host humoral immunities by targeting upstream genes in the Toll and JAK/STAT pathways...."

As the gene expression of some immune signal pathway was decreased, did these treatments affect the host immune response to foreign bacterial infection, i.e. *E. coli* or *S. aureus*?

>>Line 195. It was found that hemocytes from insects injected with miR-2989 or miR-199b-5p inhibitor were able to bind to the Sephadex A-25 beads added into the cell culture, similar to the cells from non-infected insects...

Did the authors test the specificity of the inhibitors? In Fig.4C, the encapsulation rate of inhibitor treatment group is much higher than 'cell only' group, which shows that the inhibitor may also inhibit miRNA homologous in the cells. Thus, it is necessary to find out if there are miR-2989 or miR-199b-5p homologous in the host genome.

>>Line 202. Inhibitors of miR-2989 and miR-199b-5p restore the development of parasitized *S. litura* larvae and damage the *S. manilae* eggs...

The subtitle and the results did not match here, and there were no results or descriptions about the *S. manilae* eggs in the parasitized *S. litura*.

>>Line 213. however, *S. litura* injected with miR-2989 and miR-199b-5p inhibitors were all healthy after *S. manilae* parasitism and entered into the pre-pupation phase successfully...

The results here are difficult to follow. I assume the authors hope to state that the host immunosuppression is very important for the development of *S. manilae* larvae. However, the results are not clear: (1) in Fig. 5A, the parasitism of *S. manilae* alter the host growth. It seems that the miRNA inhibitors can rescue the development arrest in the parasitized host. Do these two miRNAs also have function in development regulation? (2) what happened to those un-pupated wasp larvae? Were they attacked by the host immune system via encapsulation or phagocytosis?

>>Materials and methods

This part is too simple and miss many detail information.

>>Line 299. The resected calyxes were then homogenized using a mortar and pestle and clarified by centrifugation at 3000 rpm for 5 s.

The way the authors used to isolate PDV viral particle is too rough to eliminate the contamination of genomic DNA and ovary protein from wasps.

>>Line 305. We collected a total of 20 second-instar *S. litura*: 10 that were parasitized by *S. manilae* and 10 that were unparasitized."

Please clarify if the authors have removed wasp eggs (and teratocytes) in the parasitized larvae when constructing the small RNA library. Did the authors conduct any tests to exclude the contamination of the host?

>> Line 380. One day later, each of 80 larvae were infected with 10 *S. manilae* which were

removed after 48 h of parasitism. The number of *S. manilae* that successfully pupated was calculated; this was considered to be the viability of *S. manilae* under phagocytosis and encapsulation

There are many factors that may play roles in wasp development. Are there any evidence or literature to show that immune response of *S. litura* larvae can affect the wasp pupation rate? if not, why don't directly detect the phagocytosis and encapsulation of host after parasitism? The authors used 10 wasps to parasitize 80 host larvae, how did the authors confirm that each host larva was indeed parasitized?

Some additional comments:

>>Introduction.

To easily understand the results, please add more background information about the immune or development regulation of *Spodoptera litura* larvae by the parasitoid *S. manilae*.

>>Line 115. "A total of 113 miRNAs..."

Please attach a list that contain miRNAs and their target prediction information.

>>Line 128. 'Table S3' should be 'Table S1'?

>>Line 160. "...regulate these immune genes, probably through degradation"

How do the authors come to this conclusion? Were there any literatures described the way these two miRNAs work?

>>Line 164. "Co-transfection of these EGFP reporter plasmids..."

The transfection procedure is lost in the method section.

>>Line166. "...evaluated by fluorescence intensity (Fig. 3D) and western blotting analysis (Fig. 3E)"

In Fig. 3, Fig. 3D should be the western blotting result and Fig. 3E be the fluorescence intensity result. The western blotting result did not match the fluorescence intensity result. In Fig. 3D, the protein level of EGFP decreased about 50%, while the fluorescence intensity in Fig. 3E decreased about 20 times than the control.

>>Line 297. ...in a 26°C incubator?

>> Line 313. Target genes were predicted for miRNAs with ...

What is 'known sequences' refers to?

>>Line 325. Please change ddH2O to ddH2O

>>Line 348. Fourth-instar larvae were injected with SmBV.

What is the dosage of SmBV?

>>Line 378. Two microlitres of miRNA inhibitor

What is the concentration of miRNA inhibitors?

Response to Reviewers' Comments

Manuscript ID: COMMSBIO-19-1955-T

Title: *Snellenius manilae* bracovirus-encoded microRNAs as regulators in host immune responses

Reviewer: 1

Thank you for your constructive comments. As a result, we have performed additional experiments and modified the manuscript. Changes related to your comments are highlighted with blue in the manuscript.

Comments:

In the manuscript, "*Snellenius manilae* bracovirus-encoded microRNAs as regulators in host immune responses", Cheng-Kang Tang and co-workers proposed that *Snellenius manila* bracovirus-edcoded microRNAs (miRNAs), miR-2989 and miR-199b-5p, inhibited host immune responses via regulating host Toll and JAK/STAT pathways. The authors used Small RNA Hiseq Next Generation Sequencing combined with physiological assays to detect the functions of the two miRNAs. In general, the paper is interesting but the writing of this manuscript is not clear. Especially in the abstract, it didn't even contain the important results of this paper. Thus, I believe this manuscript is not ready for publish. Some suggestions are listed below for the authors' consideration.

Response:

We have re-written part of the abstract to include the key findings of our study (see below). Furthermore, we have made significant changes to other sections of the manuscript.

Abstract

1. To avoid inducing immune and physiological responses in their insect hosts, parasitoid wasps have developed several mechanisms to inhibit them during parasitism, including the production of venom, specialized wasp cells, and symbioses with polydnviruses (PDVs). (lines 18–21, page 2)
2. Microinjecting the inhibitors of these two miRNAs into parasitized *S. litura* larvae not only significantly decreased the pupation rate of *Snellenius manilae*, but also decreased the encapsulation activity of the hemocytes. The results demonstrate that these two SmbV-encoded miRNAs play an important role in suppressing the immune responses of parasitized hosts. Overall, our study

uncovers the functions of two SmBV-encoded miRNAs in regulating the host innate immune responses upon wasp parasitism. (lines 28–34, page 2)

1. The authors should combine Fig. 1 and 2 to make the paper more logical such that it shows the importance of the two miRNAs and the way to discover them.

Response:

Figures 1 and 2 have been combined and the corresponding parts of the manuscript have been modified.

2. It would be better to use Head Capsule Width to describe the inhibition of miRNA in Fig.5A and please also check whether the legend, "CN: *S. litura* parasitized", is correct.

Response:

We have added the head capsule width data (new Fig. 4C) in the updated manuscript. Our apologies for the error in the legend; it should be "(Control: unparasitized *S. litura*)". This has now been corrected (lines 747–748, page 21).

Fig. 4

Fig. 4 Inhibition of SmBV miRNA expression decreases *S. manilae* parasitism rate.

(A) Third-instar *S. litura* larvae were microinjected with miRNA inhibitors before parasitism by *S. manilae* wasps for 48 h. Arrow: *S. manilae*. The developments of *S. litura* larvae were measured by their (B) body sizes (lengths) and (C) head capsule width after parasitism (Control: unparasitized *S. litura*).

3. It would be helpful to add the methods used in this research to allow reviewers to understand the assays in this manuscript. For example, in Fig.3, how did the authors generate pKShE plasmid and perform western blot.

Response:

We have added new sections in materials and methods to cover pKShE plasmid generation, DNA/microRNA transfection, EGFP expression, fluorescence quantification, and Western blotting analysis (lines 368–378, page 11; lines 388–402, page 11).

“Plasmid construction

EGFP reporter plasmids pKShE-199b-5p 3'UTR and pKShE-2989 3'UTR were constructed using the plasmid pKShE in which an EGFP gene is driven by the *hsp 70* promoter in the backbone of pBluescript II KS(-) (Stratagene)^{62,63}. The 3' UTRs of *domeless* and *toll-7* were amplified by PCR from the cDNA of *S. litura* and inserted downstream to *egfp* in the plasmid pKShE to generate the plasmids pKShE-199b-5p 3'UTR and pKShE-2989 3'UTR, respectively. The primer sets and 5'-GGTTAGTACCTATATGTGAATA-3' and 5'-AGCACGTCTATTGAATGTTGCTGTC-3' were used for the 3' UTR of *domeless* and 5'-AGAGGCGTGGGCCAAACCGAC-3' and 5'-CACAAATTCCCCCTCGGCAA-3' were used to amplify the 3' UTR of *toll-7*.”

“Western blotting analysis

After reporter plasmid and miRNA mimic (10 pmol) transfection, SL1A cells (2×10^5) were placed in the wells of a 24-well plate, washed twice with PBS buffer, and lysed in RIPA lysis buffer (100 μ L). Sodium dodecyl sulfate (SDS) sample buffer was mixed with the cell samples in a 1:4 ratio. The mixture was then heated to 100°C for 10 min and then separated on a 10% SDS-polyacrylamide gel. Proteins on the gel were transferred onto a PVDF membrane (Millipore) by electroblotting for 1 h in transfer buffer. The PVDF membrane was blocked with PBS containing 5% skim milk, 0.05% Tween 20 for 1 h and incubated with primary mouse anti-EGFP (Millipore) or anti-Actin (Millipore) antibodies for another 1 h. The membrane was washed three times

with PBS containing 0.05% Tween 20 (PBST) and then incubated with secondary antibody conjugated with horseradish peroxidase (HRP) for 1 h. After three washes with PBST, the membrane was rinsed with enhanced chemiluminescence (ECL) substrate and the luminescence signal was detected using X-ray film exposure. All bands were analyzed and quantified using AlphaView SA.”

4. The authors should read this manuscript carefully to correct some wrong descriptions.

Response:

We have re-written several descriptive sections of the manuscript, as shown below, along with correcting several misspellings and mistakes.

“Introduction

1. Parasitic wasps possess many effectors to regulate host physiological mechanisms during parasitization to ensure their offspring successfully develop in the host. These include the production of venoms, symbiosis with polydnviruses (PDVs), and the release of teratocytes into the host body during egg hatching¹⁻⁵. (lines 39–42, page 2)
2. Several studies have demonstrated that PDVs can regulate the developmental functions of their hosts. The genome of TrIV, a PDV, contains a TrV family gene that has been demonstrated to suppress the proliferation of host cells, thus inhibiting host development¹⁹. In a recent study, it was found that CvBV expressed miRNAs that could inhibit ecdysone receptor (EcR) expression, resulting in delayed development of its host, *Plutella xylostella*²⁰. (lines 56 – 61, page 3)

“Results

1. The specificity between miRNA mimic and its miRNA inhibitor was demonstrated by transfecting miRNA mimic and EGFP reporter plasmid into SL1A cells with or without the corresponding miRNA inhibitor. Transfection of either miR-199b-5p or miR-2989 alone significantly decreased the EGFP level, while co-transfection of miRNA and its matching inhibitor did not have an effect on the EGFP level (Fig. S4), thus demonstrating the specificity. (lines 191–197, page 6)
2. The encapsulation activity of *S. manilae* egg and larvae was higher in both miRNA inhibitor treatment groups than in the control groups (SmBV infected or SmBV infected with inhibitor control) (Fig. 4E). These results confirmed that miRNA inhibitor treatments recovered the host cellular immune responses, which obstruct the development of *S. manilae* larvae. To further ensure that both SmBV-encoded miRNAs suppress host innate immunity rather than larval developmental pathways, we injected each miRNA mimic into second-instar larvae of *S. litura*

and assessed their development. The results showed that neither miRNAs could suppress development (Fig. 4F-H). In conclusion, our results demonstrated that inhibiting the actions of both SmBV-encoded miRNAs by miRNA inhibitors restored the host immune responses, thus suppressing *S. manilae* development in the host. (lines 222–233, page 7)

Reviewer: 2

Thank you for your constructive comments. As a result, we have performed additional experiments and modified the manuscript. Changes related to your comments are highlighted with yellow in the manuscript.

Comments:

The manuscript by Tang et al. describes microRNAs from *Snellenius manilae* bracovirus (SmBV) act as regulators in host immune responses. By comparing the miRNAs from *Spodoptera litura* larvae parasitized by *S. manila* with those unparasitized, the authors found that the SmBV encoded miRNAs into the host larvae during parasitism. Two of these miRNAs regulated the host's immune responses by blocking the expression of several key genes in the signaling pathways. Further, the authors microinjected the inhibitors of these two miRNAs into the parasitized host larvae, which rescued their immune responses and significantly decreased the *S. manila* pupation rate. The authors hope to show SmBV-derived miRNAs play an important role in regulating host immune responses in this study. However, a number of methodological issues existed in this paper, and consequently the work is not convincing. In places, the results are also overinterpreted, for example, the conclusion "Overall, we demonstrate a cross-species regulation by miRNAs in animal parasitism" in the abstract is misleading because cross-species regulation via miRNAs in animal parasitism was already discovered two years ago. I thus did not recommend acceptance of the manuscript at the present form.

Response:

To solve the methodological issues, we have performed new experiments and clarified the methods used. Furthermore, we have also modified the phrasing of several parts of the manuscript to prevent overinterpretation. We have re-written part of the abstract (see below).

Abstract

“Microinjecting the inhibitors of these two miRNAs into parasitized *S. litura* larvae not only significantly decreased the pupation rate of *Snellenius manilae*, but also decreased the encapsulation activity of the hemocytes. The results demonstrate that these two SmBV-encoded miRNAs play an important role in suppressing the immune responses of parasitized hosts. Overall, our study uncovers the functions of two SmBV-encoded miRNAs in regulating the host innate immune responses upon wasp parasitism. (lines 28–34, page 2).”

Major concerns:

>>Line 91. SmBV genome encodes abundant 91 potential miRNAs that are associated with different physiological genes in the host...”

The authors misinterpreted "the miRNAs only expressed after parasitism" the same as "SmBV-derived miRNAs". From the experiments and the method descriptions, it is very likely that the small RNA library of the parasitized *S. litura* contains miRNAs from the wasps, not only from the SmBV. Furthermore, some host miRNAs were possibly up-regulated due to SmBV viral genes or other parasitism-associated factors. Therefore it is extremely important for the authors to do additional analyses or experiments to distinguish the SmBV-derived miRNAs, wasp-derived miRNAs, and host-derived miRNAs.

Response:

We agree with this point. To distinguish the origin of miRNAs of interest, purified SmBV was directly injected into *S. litura*. The expression of the 11 miRNAs shown in Fig. 2 (new Fig. 1F and 1G) in SmBV-injected *S. litura* were analyzed by stem-loop PCR, along with those in the female wasp alone, *S. litura* parasitized by wasp, and *S. litura* not parasitized by wasp. Only samples harvested from *S. litura* injected with purified SmBV and parasitized with wasp expressed these 11 miRNAs and samples from the wasp alone did not express these 11 miRNAs. This coincides with previous publications in which no detectable SmBV transcripts were detected in the wasp (Fig. S3), thus demonstrating that these miRNAs were not wasp-derived. To further distinguish whether miR-2989 and miR-199-5p were SmBV-derived or *S. litura*-derived, a bioinformatics analysis was performed. This yielded no corresponding precursor sequences found in the genome of *S. litura*, suggesting that the miRNAs were

not *S. litura*-derived. Thus, we conclude that these miRNAs are SmBV-derived. This result has been incorporated into the manuscript (Fig. S3). (lines 141–149, page 5)

“To eliminate the possibility that these miRNAs were derived from either *S. litura* or *S. manilae*, their expression levels were analyzed in the following: *S. litura* injected with purified SmBV (Fig. S3A), female *S. manilae* alone (Fig. S3B), and both parasitized and unparasitized *S. litura* (Fig. S3C). Expression of miR-199-5p and miR-2989 was detected only in *S. litura* injected with purified SmBV or parasitized with the wasp. A bioinformatics analysis was also performed, and no corresponding precursor sequences were found on the genome of *S. litura*, confirming that miR-199b-5p and miR-2989 were derived from SmBV.”

Fig. S3. Expression levels of different miRNAs in *S. litura* infected with SmBV, female wasp, or *S. litura* with or without wasp parasitism were analyzed. The expression levels of 11 predicted miRNAs were quantified using stem-loop qPCR in *S. litura* with SmBV infection: (A), female wasp alone (B), or *S. litura* with or without wasp parasitism were analyzed (C). The expression of *let-7* miRNA serves as a positive control. At least three independent measurements were conducted for each group. A *Ct* value of 35 was set to be the cutoff for detection. Statistical analysis was

performed using Student's t-test with p values less than 0.05 (*, $P < 0.05$) to be considered statistically significant.

>>Line 102. A total of 855 known miRNAs...

The total numbers of miRNAs provided in the text are unrealistically high. No information is provided about their count numbers and pre-miRNA structures.

Response:

We have added a table as Supplementary Data 1 to list the miRNAs in *S. manila*, parasitized *S. litura*, and unparasitized *S. litura*, along with their respective count numbers. (lines 104–108, page 4)

“A total of 855 known miRNAs were detected in this analysis, of which 422 miRNAs were only expressed after parasitism (accounting for 49.4% of the total) and 173 miRNAs were exclusively expressed in uninfected hosts (accounting for 20.2% of the total) (Fig. 1A and Supplementary Data 1).”

>>Line 142-143. These results show that miR-2989 and miR-199b-5p are present in SmBV and are highly expressed after SmBV enters the host..."

Most miRNAs are very conserved across different species. Have the authors identified the existence of these two miRNAs (miR-2989 and miR-199b-5p) in the genome of the host *S. litura*? If the host genome also expresses these two miRNAs or their homologous, the conclusion will be completely changed.

Response:

We used reads containing sequences of our target miRNAs and ran a BLAST analysis against *S. litura* genomes using a bioinformatic analysis tool (Bowtie 2) to exclude the possibility that these two miRNAs were derived from *S. litura*. We further used stem loop PCR to analyze miRNA in SmBV-injected *S. litura*, female wasps alone, *S. litura* infected with wasps, and *S. litura* not infected with wasps. Only samples harvested from *S. litura* injected with purified SmBV and infected by wasps expressed the 11 target miRNAs; samples taken from wasps alone showed no miRNA expression. This is consistent with previous studies which confirmed that no SmBV transcripts were detected in wasps (Fig. S3), and shows that these miRNAs are not derived from wasps.

>>Again, in the method section, the experiment to extract SmBV is too simple to exclude the wasp ovary contents, which may lead to highly expression of these two miRNAs. Thus, the fact that the two miRNAs were highly expressed here is doubtful.

Response:

As per the above response, we have conducted stem-loop PCR and a bioinformatics analysis to confirm that both miRNAs are derived from SmBV and not from the wasps.

>>Line176. These results indicate that miRNAs produced by SmBV affect the host humoral immunities by targeting upstream genes in the Toll and JAK/STAT pathways...."

As the gene expression of some immune signal pathway was decreased, did these treatments affect the host immune response to foreign bacterial infection, i.e. E. coli or S. aureas?

Response:

We agree with your point. These two SmBV-encoded miRNAs target host *Toll-7* (Toll pathway) and *Domeless* (JAK/STAT pathway) genes; these two pathways not only regulate cellular but also humoral immune responses such as antimicrobial peptide production (Fig 1E), which has been known to play an important role in antibacterial immunity. However, cellular immunities such as hemocyte encapsulation and melanization are the main immune responses against parasitoid wasp invasion, hence we did not focus on the antibacterial response.

>>Line 195. It was found that hemocytes from insects injected with miR-2989 or miR-199b-5p inhibitor were able to bind to the Sephadex A-25 beads added into the cell culture, similar to the cells from non-infected insects...

Did the authors test the specificity of the inhibitors?

Response:

We have constructed reporter plasmids pKShE-2989 3'UTR and pKShE-199b-5p 3'UTR in which the 3'-UTR of *Toll-7* or *Domeless* were fused to the 3'-end of an EGFP coding region, respectively. This reporter plasmid and corresponding miRNA (miR-2989 and mir-199-5p) were co-transfected into SL1A cells with or without inhibitors. Transfection of either miR-2989 or miR-199b-5p alone significantly decreased EGFP

level, while co-transfection of miRNA and its matching inhibitor did not have effect on the EGFP level, demonstrating specificity between the miRNA mimic and its inhibitor. This result has now been incorporated into the manuscript (Fig. S4).

Fig. S4. The specificity between miRNA mimic and its miRNA inhibitor. Reporter plasmids pKShE-199b-5p 3'UTR (A) and pKShE-2989 3'UTR (B) in which the 3'-UTR of Domeless or Toll-7 were fused to the 3'-end of an EGFP coding region, respectively. Reporter plasmids were transfected into SL1A cells with miRNA mimic only or miRNA mimic plus its inhibitor. EGFP expression was detected at 48 hours after transfection by fluorescent microscopy.

>>In Fig.4C, the encapsulation rate of inhibitor treatment group is much higher than 'cell only' group, which shows that the inhibitor may also inhibit miRNA homologous in the cells. Thus, it is necessary to find out if there are miR-2989 or miR-199b-5p homologous in the host genome.

Response:

The statistical analysis comparing each inhibitor treatment group to the cell-only control showed no significant differences. To confirm this result and address your concern, we have conducted five more replicates. The results again displayed no significant difference between miRNA inhibitor treatments and the cell-only control. We have updated this data in the modified Fig. 3C.

Figure. Encapsulation assay determining binding of multiple hemocytes to the Sephadex A-25 beads added in the cell culture. The encapsulation rate was calculated by KP assay. All experiments were performed with eight biological replicates. Data are expressed as mean \pm standard deviation (SD). P values were determined using Student’s t-test (n.s., no significant difference).

>>Line 202. Inhibitors of miR-2989 and miR-199b-5p restore the development of parasitized *S. litura* larvae and damage the *S. manilae* eggs...

The subtitle and the results did not match here, and there were no results or descriptions about the *S. manilae* eggs in the parasitized *S. litura*.

Response:

We have changed the subtitle to “Inhibitors of miR-199b-5p and miR-2989 suppress the pupation of *S. manilae*, resulting in normal growth of parasitized *S. litura* larvae.” (lines 208–209, page 7).

>>Line 213. however, *S. litura* injected with miR-2989 and miR-199b-5p inhibitors were all healthy after *S. manilae* parasitism and entered into the pre-pupation phase successfully...

The results here are difficult to follow. I assume the authors hope to state that the host immunosuppression is very important for the development of *S. manilae* larvae.

Response:

We have re-write this section to reflect your comment (lines 222–232, page 7).

“The encapsulation activity of *S. manilae* egg and larvae was higher in both miRNA inhibitor treatment groups than in the control groups (SmBV infected or SmBV infected with inhibitor control) (Fig. 4E). These results confirmed that miRNA inhibitor treatments recovered the host cellular immune responses, which obstruct the development of *S. manilae* larvae. To further ensure that both SmBV-encoded miRNAs suppress host innate immunity rather than larval developmental pathways, we injected each miRNA mimic into second-instar larvae of *S. litura* and assessed their development. The results showed that neither miRNAs could suppress development (Fig. 4F-H). In conclusion, our results demonstrated that inhibiting the actions of both SmBV-encoded miRNAs by miRNA inhibitors restored the host immune responses, thus suppressing *S. manilae* development in the host.”

>>However, the results are not clear:

(1) in Fig. 5A, the parasitism of *S. manilae* alter the host growth. It seems that the miRNA inhibitors can rescue the development arrest in the parasitized host. Do these two miRNAs also have function in development regulation?

Response:

As the two miRNAs target genes involved in immune responses, it is more likely that they regulate immune pathways and that the rescue in development arrest is an indirect effect. The inhibited immune response generated by miRNAs when wasp eggs are laid on *S. litura* results in diapause, but injected miRNA inhibitors likely prevent immune inhibition and diapause, allowing *S. litura* to present normal immune responses to foreign pathogens and develop normally. In order to determine whether the two miRNAs regulate host development, they were injected into second-instar *S. litura* to observe any developmental changes. Our results found that injecting miRNAs caused little change in the development of *S. litura*, indicating that these miRNAs did not affect development. This result has been incorporated into the manuscript (new Fig. 4F-G) (lines 222–232, page 7).

Fig. 4

Fig. 4 Inhibition of SmBV miRNA expression decreases *S. manilae* parasitism rate. (F) Second-instar *S. litura* larvae were microinjected with miRNA mimic and monitored over a 9 days period post-injection. (Control: *S. litura* without miRNA mimic injection.) The developments of *S. litura* larvae were measured by their **(G)** body sizes (lengths) and **(H)** head capsule width after miRNA mimic injection. All experiments were performed with three biological replicates. Data are expressed as the mean and standard deviation (SD). P values were calculated using Student's t-test (***, $P < 0.005$).

>> (2) What happened to those un-pupated wasp larvae? Were they attacked by the host immune system via encapsulation or phagocytosis?

Response:

In previous studies, parasitoid wasps have been used to infect non-primary hosts, showing that hosts defend against wasp larvae through an encapsulation mechanism. Therefore, we theorized that encapsulation would also occur to the un-pupated wasp

larvae used in this study. In order to confirm this theory, we conducted encapsulation assays on wasp larvae retrieved from hosts that were parasitized with wasps. When miR-199b-5p or miR-2989 inhibitors were microinjected into the hemolymph of SmBV-infected fourth-instar *S. litura* larvae, hemocytes from the injected insects were able to bind to wasp larvae added to cell cultures in a manner similar to hemocytes collected from non-infected insects. This indicates that in hosts uninfected by SmBV or those with inhibited miR-199b-5p or miR-2989, wasp larvae will be attacked by the host's immune system via encapsulation. This result has been incorporated into the manuscript (new Fig. 4E) (lines 222–233, page 7).

Fig. 4 Inhibition of SmBV miRNA expression decreases *S. manilae* parasitism rate. (E) Encapsulation assay determining binding of multiple hemocytes to *S. manilae* eggs (upper panel) and 4-day-old *S. manilae* larvae (lower panel) added in the cell culture. Arrow: encapsulated and melanized *S. manilae* larvae. All experiments were performed with three biological replicates. Data are expressed as the mean and standard deviation (SD). P values were calculated using Student's t-test (***, $P < 0.005$).

>>Materials and methods

This part is too simple and miss many detail information.

>>Line 299. The resected calyces were then homogenized using a mortar and pestle and clarified by centrifugation at 3000 rpm for 5 s.

The way the authors used to isolate PDV viral particle is too rough to eliminate the contamination of genomic DNA and ovary protein from wasps.

Response:

We agree that our description of the extraction method is over-simplified. As such, we have added more detail to the section describing the extraction process (lines 308–326, page 10).

“*Spodoptera litura* SL1A cells were cultured in TC-100 medium (USBio) containing 10% FBS (Gibco BRL) and cultured in a 26°C incubator. SmBV virions were collected from the ovaries of female *S. manilae* wasps, as previously described³. The ovaries were dissected in pre-chilled phosphate-buffered saline (PBS), and the calyx was punctured to release the content into the PBS. The solution was filtered through a 0.45- μ m filter, and the filtrate was centrifuged at 20,000 \times g for 1 h⁵⁵. After centrifugation, the resulting pellet containing the virus was re-suspended in PBS. A plasmid containing the C fragment of SmBV was constructed to serve as the standard in qPCR to quantify the genome copy number of SmBV. The concentration of stock plasmid DNA was measured using a NanoDrop 2000 spectrophotometer (Thermo Fisher Scientific) and the amount of DNA sample was determined to be equivalent to 10¹⁰ plasmid copy number using the DNA Copy Number and Dilution Calculator (Thermo Fisher Scientific). Ten-fold serial dilution of the DNA sample ranging from 10¹⁰ to 10¹ plasmid copy number was performed, and the serially diluted samples were used as templates in qPCR. A standard curve was obtained by plotting diluted template DNA to the corresponding Ct value⁵⁶. To infect *S. litura* larvae, SmBV solution containing 10⁶ virus copy number was injected into second-instar larvae. The expression level of SmBV-encoded miRNAs was analyzed 36 h post-injection by stem-loop qPCR.”

>>Line305. We collected a total of 20 second-instar *S. litura*: 10 that were parasitized by *S. manilae* and 10 that were unparasitized."

Please clarify if the authors have removed wasp eggs (and teratocytes) in the parasitized larvae when constructing the small RNA library. Did the authors conduct any tests to exclude the contamination of the host?

Response:

We did not remove wasp eggs and teratocytes from our small RNA library, so the results of our preliminary analysis could have been polluted by hosts. In order to

eliminate these problems and ensure that our miRNA originated from the viruses, stem loop PCR was used to analyze the miRNA of *S. litura* injected with purified SmBV, female wasps alone, and *S. litura* with or without wasp parasitism. Expression of miR-2989 and miR-199-5p were detected only in *S. litura* injected with purified SmBV or those parasitized by the wasp. A bioinformatics analysis was also performed and no corresponding precursor sequences were found on the genome of *S. litura*, demonstrating that miR-2989 and miR-199b-5p were derived from SmBV.

>> Line 380. One day later, each of 80 larvae were infected with 10 *S. manilae* which were removed after 48 h of parasitism. The number of *S. manilae* that successfully pupated was calculated; this was considered to be the viability of *S. manilae* under phagocytosis and encapsulation

There are many factors that may play roles in wasp development. Are there any evidence or literature to show that immune response of *S. litura* larvae can affect the wasp pupation rate? if not, why don't directly detect the phagocytosis and encapsulation of host after parasitism?

Response:

It has previously been reported that infecting *M. sexta* parasitized by *Cotesia congregata* with other virus such as AcMNPV resulted in a decreased wasp pupation rate (Washburn et al., *J Insect Physiol* 46, 179-190 (2000)). However, there was no direct experimental evidence linking this to the host immune response. Our previous studies have shown that when energy input supporting immunity was directed towards enhancing immune responses, the pupation rate of parasitized wasps in *S. litura* exhibited significant decreases (Cheng et al., *Sci Rep* 10, 2096 (2020)). This suggests that host immunity is a contributing factor to wasp development. To respond to the reviewer's concern, we microinjected miR-2989 inhibitor or miR-199b-5p inhibitor into the hemolymph of SmBV-infected fourth-instar *S. litura* larvae. We then performed an encapsulation assay was performed to determine the binding of multiple hemocytes to foreign particles. The choice of using SmBV-infected larvae instead of wasp-parasitized larvae was to confirm that the miRNAs were indeed SmBV-derived. The encapsulation assay results showed that hemocytes from infected-insects injected with miR-2989 inhibitor or miR-199b-5p inhibitor were able to bind to the wasp larvae added into the cell culture, similar to hemocytes collected from non-infected insects (Fig. 4F-H). The hemocytes collected from SmBV-infected larvae, on the other hand,

did not exhibit encapsulation (Fig. 4E), indicating that SmBV was responsible for inducing host immune suppression.

>> The authors used 10 wasps to parasitize 80 host larvae, how did the authors confirm that each host larva was indeed parasitized?

Response:

A report published in the Taiwan Biological Prevention Measure Journal found that the parasitism rate of *S. manilae* in the field is around 91.33% (Chiu, JC and Chou, LY. 1976. Hymenopterous parasitoids of Spodoptera Litura Fab. Journal of Agricultural Research of China 25(3): 227-241 ISSN 0376-477X). Furthermore, in our experiment, *S. manilae* larvae were found in every *S. litura* we dissected after 48 hours of parasitism, confirming a near 100% parasitism rate in the host larvae.

Some additional comments:

>>Introduction.

To easily understand the results, please add more background information about the immune or development regulation of *Spodoptera litura* larvae by the parasitoid *S. manilae*.

Response:

The introduction has been modified to include more background information (lines 56–64, page 3).

“Several studies have demonstrated that PDVs can regulate the developmental functions of their hosts. The genome of TrIV, a PDV, contains a TrV family gene that has been demonstrated to suppress the proliferation of host cells, thus inhibiting host development¹⁹. In a recent study, it was found that CvBV expressed miRNAs that could inhibit ecdysone receptor (EcR) expression, resulting in delayed development of its host, *Plutella xylostella*²⁰. PDVs have previously been shown to inhibit both the cellular and humoral immunity of the host²¹. The former includes inhibition of encapsulation and a decrease in cell adhesion that inhibits phagocytosis^{22,23}, both of which result in the induction of apoptosis and, in severe cases, the disruption of hemocytes.”

>>Line 115. "A total of 113 miRNAs..."

Please attach a list that contains miRNAs and their target prediction information.

Response:

We will provide this list as Supplementary Data 1.

>>Line 128. 'Table S3' should be 'Table S1'?

Response:

We have corrected this error in the manuscript (Line 127–128, page 4).

“Overall, 11 miRNAs were selected for further study (Table S1).”

>>Line 160. "...regulate these immune genes, probably through degradation"

How do the authors come to this conclusion? Were there any literatures described the way these two miRNAs work?

Response:

We agree that there is insufficient evidence for this conclusion. As such, the sentence has been removed (lines 163–165, page 5).

“Quantitative PCR results showed that miR-199b-5p and miR-2989 inhibited the gene expression of *domeless* and *toll-7*, respectively (Fig. 2B), proving that miR-199b-5p and miR-2989 can down-regulate the expression of these immune genes.”

>>Line 164. "Co-transfection of these EGFP reporter plasmids..."

The transfection procedure is lost in the method section.

Response:

Our revised methods section now contains the procedures for plasmid construction, DNA/microRNA transfection, EGFP expression, fluorescence quantification, and Western blotting analysis (lines 368–378, page 11; lines 388–402, page 12).

“Plasmid construction

EGFP reporter plasmids pKShE-199b-5p 3'UTR and pKShE-2989 3'UTR were constructed using the plasmid pKShE in which an EGFP gene is driven by the *hsp 70* promoter in the backbone of pBluescript II KS(–) (Stratagene)^{62,63}. The 3' UTRs of *domeless* and *toll-7* were amplified by PCR from the cDNA of *S. litura* and inserted downstream to *egfp* in the plasmid pKShE to generate the plasmids pKShE-199b-5p 3'UTR and pKShE-2989 3'UTR, respectively. The primer sets and 5'-GGTTAGTACCTATATGTGAATA-3' and 5'-AGCACGTCTATTGAATGTTGCTGTC-3' were used for the 3' UTR of *domeless* and 5'-AGAGGCGTGGGCCAAACCGAC-3' and 5'-CACAAATTCCCCCTCGGCAA-3' were used to amplify the 3' UTR of *toll-7*.”

“Western blotting analysis

After reporter plasmid and miRNA mimic (10 pmol) transfection, SL1A cells (2×10^5) were placed in the wells of a 24-well plate, washed twice with PBS buffer, and lysed in RIPA lysis buffer (100 μ L). Sodium dodecyl sulfate (SDS) sample buffer was mixed with the cell samples in a 1:4 ratio. The mixture was then heated to 100°C for 10 min and then separated on a 10% SDS-polyacrylamide gel. Proteins on the gel were transferred onto a PVDF membrane (Millipore) by electroblotting for 1 h in transfer buffer. The PVDF membrane was blocked with PBS containing 5% skim milk, 0.05% Tween 20 for 1 h and incubated with primary mouse anti-EGFP (Millipore) or anti-Actin (Millipore) antibodies for another 1 h. The membrane was washed three times with PBS containing 0.05% Tween 20 (PBST) and then incubated with secondary antibody conjugated with horseradish peroxidase (HRP) for 1 h. After three washes with PBST, the membrane was rinsed with enhanced chemiluminescence (ECL) substrate and the luminescence signal was detected using X-ray film exposure. All bands were analyzed and quantified using AlphaView SA.”

>>Line166. ...evaluated by fluorescence intensity (Fig. 3D) and western blotting analysis (Fig. 3E)"

In Fig. 3, Fig. 3D should be the western blotting result and Fig. 3E be the fluorescence intensity result. The western blotting result did not match the fluorescence intensity result. In Fig. 3D, the protein level of EGFP decreased about 50%, while the fluorescence intensity in Fig. 3E decreased about 20 times than the control.

Response:

These errors have now been corrected. We agree your point. The fluorescence intensity result in Fig. 3E was based on the images analysis, so it is not as accurate as western blotting. Hence, we re-analyzed the western blotting results from three independent films and quantified the EGFP protein levels shown in (new Fig. 2D). The fluorescence image will be used for supporting the western blotting results.

>>Line 297. ...in a 26°C incubator?

Response:

This has been modified accordingly (lines 308–309, page 10).

“*Spodoptera litura* SL1A cells were cultured in TC-100 medium (USBio) containing 10% FBS (Gibco BRL) and cultured in a 26°C incubator.”

>> Line 313. Target genes were predicted for miRNAs with ...

What is 'known sequences' refers to?

Response:

The term "known sequences" has been removed as the sequences of all the identified miRNA candidates are known (lines 337–338, page 11).

“Target genes were predicted for miRNAs using RNAhybrid, miRanda, TargetScan, and PITA.”

>>Line 325. Please change ddH2O to ddH2O

Response:

This has been modified (line 349, page 11).

“and reverse primers, 1 μ L of cDNA, and 7 μ L of ddH₂O.”

>>Line 348. Fourth-instar larvae were injected with SmBV.

What is the dosage of SmBV?

Response:

This has been modified (line 425, page 13).

“Third-instar larvae were injected with SmBV (1 x 10⁶ virus copy number).”

>>Line 378. Two microlitres of miRNA inhibitor

What is the concentration of miRNA inhibitors?

Response:

This has been modified (line 405, page 13).

“Two microliters of miRNA inhibitor (10 pmol) was”

Reviewers' comments:

Reviewer #1 (Remarks to the Author):

Authors have addressed my questions.

Reviewer #2 (Remarks to the Author):

The authors did a lot of work on the last version of manuscript and the revised manuscript has addressed some of my concerns. However, several major issues still exist in the manuscript. So, I suggest "rejection" to this manuscript.

1. Line 105. Based on the Supplementary data 1, the number 855 is not correct. Some of the miRNA names are repeated. For example, there are five let-7 with different lettered suffixes in the table. However, without genomic analysis and miRNA precursor information, it is impossible to say how many miRNA genes in the genome. Besides, the miRNA reads number in Supplementary data 1 did not coincide with the result of Fig. S3A. The authors must reanalyze the small RNA Hiseq next-generation sequencing results according to the *S. litura* genome data.
2. Line 152. Though the relationship between SmBV and MdBV is very close, the total number of proviral loci and the number of proviral segments per locus differ between species. Thus, it is essential to verify the two miRNA precursor sequences in SmBV proviral genome.
3. Line 190. Did the miR-199b-5p and miR-2989 mimics inhibit encapsulation and phagocytosis of *S. litura* hemocyte? This is the most direct evidence for the authors' key point "bracovirus-encoded miRNAs function in host immune responses". As shown in Fig.3, the miRNA inhibitor works too well that almost 100% recover the immune suppression of SmBV, which is illogical because many PDV viral proteins have been shown to inhibit encapsulation and phagocytosis in host. Thus, the specificity of the miRNA inhibitor is doubtful as I pointed out in my last review.

Response to Comments

Manuscript ID: COMMSBIO-19-1955A

Title: *Snellenius manilae* bracovirus-encoded microRNAs as regulators in host immune responses

Thank you for your constructive comments. As a result, we have performed additional experiments and modified the manuscript. Changes related to your comments are highlighted with yellow in the manuscript.

Comments from the Editor:

Your manuscript entitled "*Snellenius manilae* bracovirus-encoded microRNAs as regulators in host immune responses" has now been seen by 2 referees. You will see from their comments below that while they find your work of interest, some important points are raised. We are still interested in the possibility of publishing your study in Communications Biology, but would like to consider your response to these concerns in the form of a revised manuscript before we make a final decision on publication.

While we appreciate that you have addressed all major issues we previously requested, reviewer #2 pointed out some discrepancy between Supplementary Data 1 and Supplementary Fig 3a and requested that you verify the two miRNA precursor sequences in SmBV proviral genome. We ask you to please address these remaining concerns during the last round of revision.

Response:

Thank you for your positive feedback. We also appreciate the careful review and constructive suggestions from the reviewers. In response to the specific concerns of reviewer #2, we have made several changes to the manuscript and its supplementary material, detailed below.

First, the apparent discrepancy between Supplementary Data 1 and Supplementary Fig 3a (we think that the reviewer actually means to refer to Supplementary Fig 2B or 2C rather than 3a) is the result of inadequate explanation. The reads number in Supplementary Fig 2B/2C refers to the raw reads of the next-generation sequencing, whereas the number (855) in Supplementary Data 1 reflects the number of reads that had passed quality controls and were mapped to known miRNAs. To clarify the difference between these figures, we have rewritten the description in both the manuscript and legends of Supplementary Fig 2.

Secondly, we appreciate their concern that the miRNA precursor sequences in the

SmBV proviral genome should be verified. Indeed, we would like to have met this requirement; however, there is no SmBV genome available for sequence mapping. As such, we tried several alternative methods to prove that the two miRNAs were derived from SmBV but not from the host genome. These methods included sequence mapping to the host genome and Northern blotting assays probing the precursors and mature sequences in the virus-injected samples (Supplementary Fig 3D).

Finally, we responded to the third comment of reviewer #2 by performing two different sets of experiments (results are provided in the more detailed responses below).

We sincerely hope that we have responded to all the questions and concerns raised by the reviewer. Thank you for your time and consideration.

Reviewer: 2

Comments:

The authors did a lot of work on the last version of manuscript and the revised manuscript has addressed some of my concerns. However, several major issues still exist in the manuscript. So, I suggest “rejection” to this manuscript.

1. Line 105. Based on the Supplementary data 1, the number 855 is not correct. Some of the miRNA names are repeated. For example, there are five let-7 with different lettered suffixes in the table. However, without genomic analysis and miRNA precursor information, it is impossible to say how many miRNA genes in the genome. Besides, the miRNA reads number in Supplementary data 1 did not coincide with the result of Fig. S3A. The authors must reanalyze the small RNA Hiseq next-generation sequencing results according to the *S. litura* genome data.

Response:

In our small RNA Hiseq analysis, we aligned the obtained sequences against known miRNAs. According to miRNA nomenclature, if two miRNAs have the same name but have different suffix letters (e.g., hsa-mir-451a and hsa-mir-451b), they belong to the same family but produce different mature miRNA products. The number 855 is correct as we counted all mapped mature miRNAs in the databases. Indeed, this prediction does not represent the actual number of expressed miRNAs; therefore, we

performed stem-loop qPCRs to validate the expression of miRNAs in *S. litura* larvae after *S. manilae* parasitism (Fig. 1F) and in SL1A cells after SmBV infection (Fig. 1G). The number of miRNA targets was then reduced to two (i.e., miR-199b-5p and miR-2989).

The miRNA reads number in Fig. S2B/C (we believe you are referring to this rather than S3A) does not coincide with Supplementary Data 1 because it displays raw reads data rather than clean reads. After removing sequence redundancy, we used the clean reads (Fig. S2C) to align the sequences in databases and obtained the results of Supplementary Data 1. To avoid any misunderstanding, we have modified the description in Result (lines 107–111) to the following: “Clean miRNA reads could be mapped to 855 known mature miRNAs in the databases, of which...”. Furthermore, we have also corrected the figure legend of Fig. S2.

2. Line 152. Though the relationship between SmBV and MdBV is very close, the total number of proviral loci and the number of proviral segments per locus differ between species. Thus, it is essential to verify the two miRNA precursor sequences in SmBV proviral genome.

Response:

We agree that it is important to verify the existence of the two miRNA precursor sequences in the SmBV genome. Unfortunately, there is no SmBV proviral genome available to date. As such, we tried an alternative method to prove that these two miRNAs are indeed derived from SmBV, namely a bioinformatics analysis to map the precursor sequences of miR-199b-5p and miR-2989 in the genome of *S. litura*. Since the corresponding precursor sequences could not be identified in the *S. litura* genome, these two miRNAs should be derived from the virus. The stem-loop qPCR results also support this, since the two miRNAs were only detected in *S. litura* injected with purified SmBV (Fig. S3A) or *S. litura* parasitized by the wasp (Fig. S3C), but not in female *S. manilae* alone (Fig. S3B) or unparasitized *S. litura* (Fig. S3C).

For further confirmation, we performed Northern blotting assays probing the precursors and mature sequences in samples with or without parasitism, or with virus injection. The Northern blotting results showed clear expression of both mature and precursor miRNAs in infected and parasitized samples, but not in non-parasitized *S. litura* (Fig. S3D). These results indicate that miR-199b-5p and miR-2989 were

processed from SmBV and were highly expressed only after SmBV entered the host.

Fig. S3. Expression levels of different miRNAs in *S. litura* infected with SmBV, female wasp, or *S. litura* with or without wasp parasitism. (D) Small RNAs harvested from *S. litura* with or without wasp parasitism or *S. litura* infected with SmBV were analyzed by Northern blotting using probes against mature and precursors miRNA (top panels). *let-7a* miRNA signal served as a positive control (bottom panels). 1: *S. litura* only; 2: *S. litura* with wasp parasitism; 3: *S. litura* infected with SmBV. At least three independent measurements were conducted for each group. A *Ct* value of 35 was set to be the cutoff for detection. Statistical analysis was performed using Student's t-test with *p*-values less than 0.05 (*, *p* < 0.05) considered statistically significant.

3-1. Line 190. Did the miR-199b-5p and miR-2989 mimics inhibit encapsulation and phagocytosis of *S. litura* hemocyte? This is the most direct evidence for the authors' key point "bracovirus-encoded miRNAs function in host immune responses".

Response:

In response to your comment, we microinjected miR-199b-5p and miR-2989 mimics into *S. litura* and discovered that there was indeed an inhibition effect on

encapsulation and phagocytosis of *S. litura* hemocytes. This, in turn, affected host immune responses to foreign bacterial infection. Our new results showed that *E. coli* infection following the injection of these two mimics caused mortality rates to rise significantly in *S. litura*, indicating that these miRNA mimics can inhibit immune responses and increase susceptibility to pathogens. We have incorporated these results into the updated version of our manuscript. (lines 224–243, page 7; lines 818–833, page 23)

“To further ensure that these two SmBV-encoded miRNAs function to suppress host innate immunity, miR-199b-5p mimic or miR-2989 mimic were injected into third-instar larvae of *S. litura* to assess their effects on phagocytosis and encapsulation. Gene expression analysis result showed that the expression of immune genes in the JAK/STAT pathway and the Toll pathway were significantly decreased after miRNA mimics injection (Fig. S6). This in turn contributed to the suppressed phagocytic and encapsulation activity in insects injected with miR-199b-5p mimic or miR-2989 mimic (Fig. 4A and 4B). To assess the effect of these two miRNA mimics on host immune responses to foreign pathogens, 24 h after the microinjection of miR-199b-5p mimic or miR-2989 mimic, *E. coli* K-12 strain (1×10^5 colony-forming unit (CFU)) was injected into third-instar *S. litura* larvae. The survival rate of injected insects was monitored every hour for up to 144 h (Fig. 4C). A *p*-value of less than 0.05 was considered statistically significant. Prior injection of either miR-199b-5p mimic ($p < 0.001$) or miR-2989 mimic ($p < 0.01$) significantly decreased the survival rate of *S. litura* after infection with *E. coli* K-12 as compared to *S. litura* not injected with miRNA mimic or injected with control mimic (Fig. 4C). At 144 h, the survival rate of control mimic larvae was 50%, while that of larvae injected with either miRNA mimics was less than 10%. These results demonstrate that miR-199b-5p and miR-2989 can inhibit immune responses against foreign pathogens and significantly increase mortality from *E. coli* infection.”

Fig. 4 SmBV-encoded miR-199b-5p and miR-2989 suppress cellular immune responses in *S. litura*. Third-instar *S. litura* larvae were microinjected with SmBV, miR-199b-5p or miR-2989 mimics. **(A)** Phagocytosis activity of injected *S. litura* larvae; green fluorescence is emitted from the ingested *E. coli* by hemocytes. The phagocytosis ratio (%) was derived from the ratio of FITC to DAPI. **(B)** Encapsulation assay showing binding of multiple hemocytes to Sephadex A-25 beads added to the cell culture. The encapsulation rate was calculated by KP assay. Bottom: representative images of the Sephadex A-25 beads added to each cell culture. **(C)** Survival rate of larvae in response to infection by *E. coli* K12. Kaplan-Meier survival curve with log-rank test (Wilcoxon-Breslow-Gehan Test) comparing survival of *S. litura* larva infected with *E. coli* K12. A *p*-value of less than 0.05 was considered statistically significant. Pairwise comparison: with NC mimic vs. with miR-199b-5p, $p < 0.0001$; with NC mimic vs. with miR-2989, $p < 0.01$. NC mimic: negative control mimic. All experiments were performed with five biological replicates. Data are expressed as the mean and standard deviation (SD). *p*-value were calculated using Student's *t*-test (*, $p < 0.05$; **, $p < 0.01$; ***, $p < 0.0005$).

3-2. As shown in Fig.3, the miRNA inhibitor works too well that almost 100% recover the immune suppression of SmBV, which is illogical because many PDV viral proteins have been shown to inhibit encapsulation and phagocytosis in host. Thus, the specificity of the miRNA inhibitor is doubtful as I pointed out in my last review.

Response:

It has been reported that infection by viral pathogens typically enhances cellular immune response in *Lepidoptera* by 2–3 fold as a means of eliminating the pathogens (Buyukguzel, *et al.*, *Journal of Insect Physiology*. 2007. Vol 53, p99–105). Furthermore, our previous study showed that phagocytosis activity was enhanced in *S. litura* after infection by baculovirus AcMNPV and was suppressed during the infection of SmBV with the same titer (Chang. *et al.*, *Scientific Reports*. 2020 Feb 7;10(1):2096). In the present study, the microinjection of miRNA inhibitors restored the phagocytosis activity of SmBV-infected hosts to levels similar to that of control/uninfected hosts. However, this restored level of phagocytosis activity may still be lower than that in hosts infected with AcMNPV, for which there is no known immune suppression mechanism in *S. litura* and only results in the normal immune response, as per our previous results. As such, the addition of miRNA inhibitors could not fully restore the phagocytosis activity to the level caused by other pathogens because PDV viral proteins (from the added SmBV) still inhibited encapsulation and phagocytosis in the assayed *S. litura*.

In order to verify our results, we repeated the experiments with twice the number of larvae in each group compared with our original study and similar results were obtained. To account for observational errors in fluorescence microscopy, we used flow cytometry to analyze the phagocytic activity of hemocytes; the updated results were similar to those obtained with fluorescence microscopy (new Fig. S5 in the updated manuscript; lines 211–213, page 7). Additionally, we conducted an *in vivo* encapsulation assay in which *S. litura* larvae were injected with nematodes; the results were consistent with our original *in vitro* study. These indicate that our results are highly reproducible (new Fig. 3D in the updated manuscript; lines 217–222, page 7). All these results demonstrate that injection of miRNA inhibitors can rescue the immune suppression induced by SmBV.

To clearly describe our findings, we modified the corresponding descriptions in the manuscript: (1) “The phagocytosis assay result showed that phagocytosis capacity was restored in insects microinjected with miR-199b-5p inhibitor or miR-2989 inhibitor compared to insects injected with negative control inhibitor or solely with virus

infection (Fig. 3A-B)” (lines: 209-212); (2) “It was found that hemocytes from insects injected with miR-199b-5p inhibitor or miR-2989 inhibitor increased the binding to Sephadex A-25 beads added into the cell culture, compared to hemocytes added with NC inhibitor or with virus infection (Fig. 3C)” (lines 215–218).

Fig. S5. Inhibition of SmBV miRNA expression increases phagocytosis responses in *S. litura*. Third-instar *S. litura* larvae were injected with SmBV, miR-199b-5p or miR-2989 inhibitor and hemocytes were extracted 36 h after injection. The hemocytes were mixed with pHrodo™ Red dye conjugated *E. coli* and the phagocytosis activity of hemocytes was determined by flow cytometry. **(A)** The percentage of phagocytic cells (%) was derived from the proportion of pHrodo™ Red dye conjugated hemocytes. **(B)** Upper: representative histograms showing the percentage of red-fluorescence-positive cells from the phagocytosis activity of hemocytes. Lower: labeled *E. coli* and hemocytes were observed using fluorescence microscopy. All experiments were performed with four biological replicates. Data are expressed as the mean \pm standard deviation (SD). *p*-values were calculated using Student’s *t*-test (*, *p* < 0.05).

Fig. 3 Inhibition of SmBV miRNA expression increases cellular immune responses in *S. litura*. (D) Encapsulation of *C. elegans* in the hemocoel of *S. litura*. The encapsulated nematodes recovered from *S. litura* at 24 h after injection. NC inhibitor: negative control inhibitor. All experiments were performed with eight biological replicates. Data are expressed as the mean and standard deviation (SD). *p*-values were calculated using Student's t-test (*, $p < 0.05$; **, $p < 0.01$; ***, $p < 0.0005$).

REVIEWERS' COMMENTS:

Reviewer #2:

Remarks to the Author:

The revised manuscript did not address my concerns. As for the response of 2, the information of high-quality genomes of both the parasitoid *Snellenius manilae* and its associated bracovirus SmBV are the base for accurate identification of miRNAs, so the authors are strongly required to verify their results with the genome data of both the parasitoid and its associated bracovirus. As for the response of 3-2, the authors stated that "However, this restored level of phagocytosis activity may still be lower than that in hosts infected with AcMNPV, for which there is no known immune suppression mechanism in *S. litura* and only results in the normal immune response...", It seems that the authors compared the effect of miRNA inhibitors on phagocytosis activity with that in hosts infected with AcMNPV. Why? I didn't see any connection here. It is totally illogical! The authors claimed that their results are highly reproducible, I suggest testing the expression of their target genes or their protein level post miRNA inhibitor treatment to build the connection between the expression of target genes and phagocytosis in *S. litura*.